# Empowering Efficiency and Efficacy in WebAgent via Enabling Info-Rich Seeking

**Zhengwei Tao**[1,3,4,*]   **Haiyang Shen**[1,4,*]   **Baixuan Li**[2,4,*]   **Wenbiao Yin**[4]   **Jialong Wu**[1,2,4]

**Kuan Li**[4]   **Zhongwang Zhang**[4]   **Huifeng Yin**[4]   **Rui Ye**[4]   **Yong Jiang**[4]

**Pengjun Xie**[4]   **Fei Huang**[4]   **Jingren Zhou**[4]   **Wentao Zhang**[1,3,†]   **Yun Ma**[1,†]   **Zhiqiang Gao**[2,†]

[1]Peking University   [2]Southeast University   [3]Zhongguancun Academy   [4]Tongyi Lab, Alibaba Group

wentao.zhang@pku.edu.cn   mayun@pku.edu.cn   zqgao@seu.edu.cn

[*]Equal Contribution   [†]Corresponding Author

## Abstract

Large Language Model (LLM)-based agents have emerged as a transformative approach for open-ended problem solving, with information seeking (IS) being a core capability that enables autonomous reasoning and decision-making. While prior research has largely focused on improving retrieval depth, we observe that current IS agents often suffer from *low search efficiency*, which in turn constrains overall performance. A key factor underlying this inefficiency is the sparsity of target entities in training tasks, which limits opportunities for agents to learn and generalize efficient search behaviors. To address these challenges, we propose `WebLeaper`, a framework for constructing high-coverage IS tasks and generating efficient solution trajectories. We formulate IS as a tree-structured reasoning problem, enabling a substantially larger set of target entities to be embedded within a constrained context. Leveraging curated Wikipedia tables, we propose three variants for synthesizing IS tasks—`Basic`, `Union`, and `Reverse-Union`—to systematically increase both IS efficiency and efficacy. Finally, we curate training trajectories by retaining only those that are simultaneously accurate and efficient, ensuring that the model is optimized for both correctness and search performance. Extensive experiments conducted on five IS benchmarks—BrowserComp, GAIA, Seal-0, WideSearch, and xbench-DeepSearch—demonstrate that our method consistently achieves improvements in both effectiveness and efficiency over strong baselines.

## 1 Introduction

The LLM-based agents mark a paradigm shift in AI, delivering transformative solutions to challenges once deemed intractable across diverse domains (Guo et al., 2024; Ye et al., 2023). Among their core capabilities, information seeking (IS) plays a crucial role in enabling the cognitive autonomy of these agents. This ability not only drives their adaptability in open-ended tasks but also underpins a new generation of powerful commercial systems, including OpenAI Deep Research (OpenAI, 2025b), Google's Gemini (Gemini, 2025), and Perplexity AI (Perplexity, 2025), Kimi-Researcher (Team, 2025).

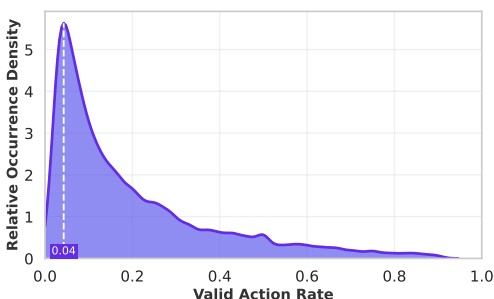

Figure 1: The distribution of valid actions of the agent based on the GPT model on our synthesized IS task. The valid actions are those seeking the correct target entities required by the question.

While numerous studies have sought to enhance the IS capabilities of agents through complex question–answering pipelines and advanced fine-tuning strategies (Wu et al., 2025a; Li et al., 2025c;b; Tao et al., 2025; Qiao et al., 2025; Lu et al., 2025), most existing approaches primarily concentrate on improving the search depth, giving comparatively little attention to search efficiency. Our preliminary experiments indicate that current LLM-based agents search inefficiently. As shown in Figure 1, the distribution of valid actions for a competitive IS agent peaks around 0.04, meaning that in most cases, only a small fraction of actions are effective (Wong et al., 2025; Xue et al., 2025). This low valid-action rate reflects suboptimal search behaviors, including redundant query reformulations,

retrieval of irrelevant information, and unnecessarily long search chains. Such inefficiencies not only increase computational and time costs but also limit the agent's overall IS performance.

The design of synthetic training tasks incurs this inefficiency. In typical IS agent setups, the agent begins with a set of known entities and incrementally gathers information to infer all target entities. However, prior work often constructs tasks in which the target entities are overly sparse (Wu et al., 2025a; Li et al., 2025c;b). Such sparsity limits the agent's exposure to informative cues, reducing opportunities to learn to locate relevant information within a constrained context window. As a result, the agent spends more actions processing irrelevant content, weakening its search strategies, leading to lower performance. Furthermore, it can bias the measurement of search efficiency, which we prove in a later section. This bias makes it difficult to obtain an accurate training signal, thereby obstructing the systematic learning of more efficient search behaviors. These limitations underscore the need to redesign training tasks, enabling optimized seeking efficiency and stronger IS capabilities.

To address these challenges, we propose `WebLeaper`, a framework designed with two core objectives: (1) to construct new IS tasks containing a substantially larger number of target entities; and (2) to generate solution trajectories that achieve both high accuracy and high efficiency. For the first objective, we model the IS process as a tree-structured reasoning task, which compactly accommodates more target nodes within a limited context. Based on this formulation, we systematically increase task complexity through three dataset variants. First, leveraging curated Wikipedia tables, we synthesize **Basic** version, which directly addresses the challenge of entity sparsity by creating a high-density search space within a single, structured source. To mirror more realistic scenarios that demand integrating information from multiple sources, our **Union** variant constructs tasks that require synthesizing facts across different sources, thereby increasing search ambiguity. Finally, to mitigate the risk of agents adopting simplistic, keyword-based shortcuts, the **Reverse-Union** variant reverses the logical flow, compelling the agent to first deduce intermediate entities from scattered clues before completing the main search task. For the second objective, we construct task-completion trajectories that are filtered according to *Information-Seeking Rate* (ISR) and *Information-Seeking Efficiency* (ISE), retaining only those that solve the task both accurately and efficiently. Models trained on this curated dataset yield our final IS agent.

We conduct extensive experiments to evaluate our approach across five benchmarks: Browser-Comp (Wei et al., 2025), GAIA (Mialon et al., 2023), Seal-0 (Pham et al., 2025), WideSearch (Wong et al., 2025), and xbench-DeepSearch (Xbench-Team, 2025). Our method achieves consistent improvements on all benchmarks. Ablation studies on the dataset design further confirm the effectiveness of our proposed components. We summarize our contribution as follows:

- We design a new information-seeking task formulation on a tree-structured reasoning problem, leading to the inclusion of a substantially larger set of target entities within a constrained context. Based on this formulation, we construct the *Basic*, *Union*, and *Reverse-Union* datasets.
- We generate and filter task-solving trajectories using the proposed *Information-Seeking Rate* (ISR) and *Information-Seeking Efficiency* (ISE) metrics, retaining only those trajectories that solve tasks both accurately and efficiently.
- We conduct extensive experiments on five public IS benchmarks, BrowserComp, GAIA, Seal-0, WideSearch, and Xbench-DeepSearch, achieving consistent improvements over strong baselines.

## 2 DEFINITIONS

An Information-Seeking (IS) task challenges an agent to answer a complex natural language question by navigating a vast information space to assemble a complete set of required entities. This process is inherently sequential, involving the progressive discovery of entities, understanding their properties (attributes), and leveraging relationships between them to uncover further entities. This section formally defines the components of such a task and the metrics for evaluating an agent's performance, emphasizing the importance of identifying both final and intermediate entities in the reasoning chain.

### 2.1 INFORMATION-SEEKING TASK

An entity $e \in \mathcal{E}$ is the fundamental unit of information. An *Information-Seeking (IS) task* is the process of identifying and collecting a specific set of target entities from $\mathcal{E}$, based on a question. Formally, an IS task is a tuple: $\mathcal{T} = \langle q, R \rangle$, where $q$ is the natural language question and $R \subset \mathcal{E}$ is the set of the target entities that collectively satisfy the conditions posed by $q$.

Critically, the required set $R$ includes not only the final, explicit answers but also all *intermediate entities* that are necessary stepping stones in the reasoning process. Consider the question:

$$q : \textit{Which player of a team in the 2004–05 season, who was born in the 1990s?}$$
$$\textit{This team was founded in 1966 and is an East German football team.} \tag{1}$$

To solve this, an IS agent must seek for information online, and find the target entity set as answer:

$$R = \{\textit{Robert Rudwaleit}, \textit{Danny Kukulies}, \ldots\}. \tag{2}$$

## 2.2 INFORMATION-SEEKING AGENT

We focus on an *Information-Seeking Agent* that interacts with a web environment to solve an IS task $\mathcal{T}$ within the ReAct framework (Yao et al., 2023). The agent's operation is a sequential decision-making process occurring over discrete time steps $t = 1, \ldots, T$. At each step, the agent analyzes its current state (including the initial question and all previously gathered information), generates a thought for planning its next move, executes a tool-based action to seek new information, and receives an observation from the environment. This entire process is captured in the *agent trajectory* is defined as

$$\mathcal{H}_T = (q, \tau_1, \alpha_1, o_1, \tau_2, \alpha_2, o_2, \ldots, \tau_T, \alpha_T, o_T), \tag{3}$$

where $\tau_i$ is the planning thought, $\alpha_i$ is the seeking action, and $o_i$ is the resulting observation at step $i$. At the end of the process, the agent has obtained a set of entities $O \subset \mathcal{E}$, which is the union of all unique entities discovered across all steps.

## 2.3 QUANTIFYING INFORMATION COLLECTION AND EFFICIENCY

To guide an agent towards successfully solving IS tasks, its performance framework must value the entire reasoning process, not merely the final output. Our central thesis is that by explicitly quantifying the value of *all* required information discovered, we can create a stronger signal for learning effective search strategies. To this end, we define principles to formalize the performance (the total information gain) and the efficiency (the gain per action) of the agent's collection process.

**Information-Seeking Rate (ISR)** Recall that $R$ denotes the set of target ground-truth entities for the task, with cardinality $n = |R|$. $O$ is the set of entities actually obtained by the agent during its operation. The intersection $R \cap O$ therefore contains all required entities that were successfully retrieved. The *information collection rate* directly measures the fraction of required entities successfully obtained by the agent:

$$\text{ISR} = \frac{|R \cap O|}{|R|} = \frac{|R \cap O|}{n}. \tag{4}$$

$\text{ISR} \in [0, 1]$, and higher values indicate more thorough coverage of the required information.

**Information-Seeking Efficiency (ISE)** While ISR measures completeness, the *information collection efficiency* reflects the average number of action steps to discover the target entity:

$$\text{ISE} = \frac{n}{T}, \tag{5}$$

where $T$ is the total number of steps of the solving trajectory. Higher ISE implies greater IS efficiency. The stability of measuring ISE is important for providing unbiased training signals.

**Proposition 1** (Variance of ISE). *Let $X_i$ denote the number of steps the agent takes to discover the $i$-th new entity in $R$. Therefore $\text{ISE} = \frac{n}{T} = \frac{n}{\sum_{i=1}^{n} X_i}$. Assume $X_1, \ldots, X_n$ be i.i.d. random variables with finite mean $\mu > 0$ and finite variance $\sigma^2$, $X_i > 0$ almost surely, then:*

$$\text{Var(ISE)} = \mathcal{O}\left(\frac{1}{n}\right). \tag{6}$$

This proposition shows that as the number of target entities $n$ grows, measuring ISE becomes a more stable and reliable performance metric. The detailed proof is provided in Appendix A.2.

## 3 METHOD

To enhance the information efficiency of the IS agent, our approach trains the model on a calibrated task $\mathcal{T} = \langle q, R \rangle$ together with the corresponding task-solving trajectory $\mathcal{H}$. In prior IS agent training setups, the dataset typically contained only a limited number of target entities ($R$). This design substantially restricts the potential improvement in information-seeking efficiency and, in turn, limits the agent's overall capability. The limitation incurs two problems:

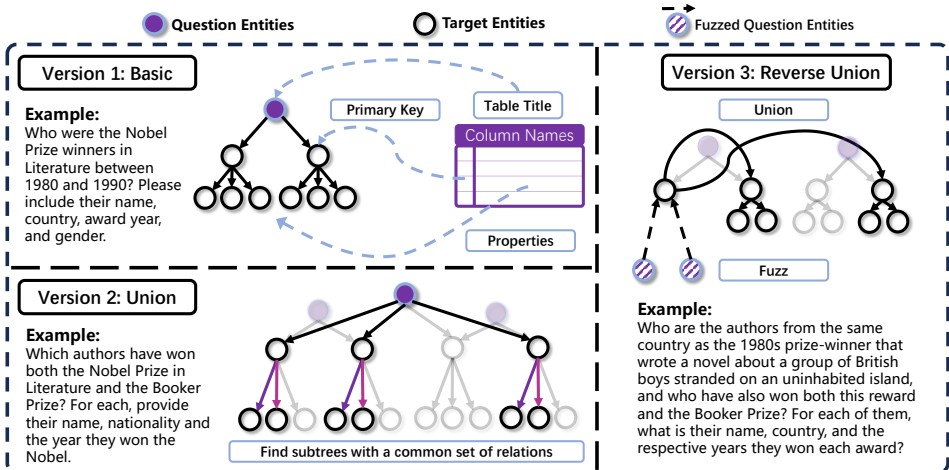

Figure 2: An overview of `WebLeaper`. The reasoning structure is modeled as a tree. A root entity (question entity) connects to a set of second-layer entities. **(a) Version-I (Basic)** constructs a simple reasoning tree from a single information source. **(b) Version-II (Union)** creates a complex task by finding a maximal union between two trees that share a common set of relations within their subtrees (e.g., both have has_nationality). **(c) Version-III (Reverse-Union)** reverses the reasoning process. It provides fuzzed clues (third-layer entities) as question entities, forcing the agent to first deduce a second-layer anchor entity (an entity from the second layer), then other relevant subtrees.

- With a small volume of $R$, it is difficult to train the agent to retrieve information efficiently within a limited context length.
- Our method relies on measuring the information-seeking efficiency ISE. As shown in Eq. (6), a small set of target entities introduces measurement bias in the ISE metric.

To overcome these shortcomings, we introduce `WebLeaper`, a novel data synthesis framework specifically designed to boost information-seeking efficiency. Our method consists of two main components: (1) a QA synthesis pipeline for generating calibrated tasks, and (2) a trajectory construction process for producing realistic task-solving sequences. We describe the QA synthesis pipeline and trajectory construction process in detail in the following subsections. For detailed walkthroughs of the examples for each synthesis version, please refer to Appendix A.8.

## 3.1 Entity-Intensive Task Synthesis

### 3.1.1 Version-I: Basic

In an information-seeking task, the reasoning structure matters. We use a tree, denoted as $T_i$, to represent this structure, where nodes are entities and edges are relations between them. The IS agent must start with some known entities in the tree and reason along the edges to determine the target ones. To incorporate as many target entities as possible, we use this tree structure for its compact and hierarchical organization.

Synthesizing such a task $\mathcal{T} = \langle q, R \rangle$ requires a large volume of relevant entities, which is non-trivial. Following the one-entity-at-a-time collection strategy of prior work is prohibitively expensive. Therefore, we exploit the structured tables contained in Wikipedia articles, which encapsulate rich relational information. These tables naturally provide groups of entities connected by specific relationships, enabling us to efficiently construct the reasoning tree $T_i$. We crawled approximately 2 million tables from Wikipedia and applied a multi-stage cleaning procedure, retaining only large, well-formed, and structurally homogeneous tables. The detailed data cleaning procedure and construction rationale are described in Appendix A.6.

To construct the reasoning structure illustrated in Figure 2(a), we populate its layers using information from a single table. The entities extracted from the table title form the root of the tree (i.e., the question entities). Next, we employ an LLM to select the most representative, non-redundant column of values from the table—typically the primary key—as the second-layer entities (e.g., 'Czesław Miłosz'). An edge between the root entity and a second-layer entity indicates that the table contains this entity. The third-layer entities are derived from the remaining columns of the table, with their

values representing attributes of the corresponding second-layer entity (e.g., 'country: Poland', 'year: 1980'). In this layer, an edge signifies that the second-layer entity possesses the given property defined by the third-layer entity.

Each second-layer entity and its associated third-layer entities form a subtree, which we denote as $S_{i,j}$. These subtrees, each possessing a set of relations $\text{Rel}(S_{i,j})$ that connect its layers, represent cohesive units of information (e.g., a specific laureate and all their details). The full reasoning tree $T_i$ is thus composed of a set of such subtrees $\{S_{i,j}\}$. The question provides the root entities, while all entities in the subtrees (both second and third layers) constitute the final answer. The detailed construction process and the required reasoning path for the example task are explained in Appendix A.8.1.

Notably, we use Wikipedia primarily for two reasons. First, it is a large, comprehensive, widely-used encyclopedic resource with broad domain coverage. Second, its semi-structured tables offer rich, clean, reliable relational data—ideal for entity-intensive task synthesis—with sufficient quality and diversity to build various complex meaningful tasks.

### 3.1.2 VERSION-II: Union

While effective, the reasoning structure of our basic tasks is derived from single sources, limiting their structural complexity and the scope of questions we can pose. To address this, we aim to construct tasks with a more intricate reasoning structure that spans multiple information sources by uniting reasoning trees from our Basic version that share similar themes and structures.

To generate more challenging questions, we propose uniting reasoning subtrees in Basic version that share similar themes and structures. A naive approach, such as randomly combining subtrees, often results in semantically incoherent questions. To systematically discover the most substantial integration opportunities, our approach models this as a Union operation, which identifies multiple reasoning trees whose respective subtrees share some common relations.

The primary challenge is to systematically search the entire collection of trees to find all groups that are suitable for union. To avoid a combinatorial explosion from enumerating all possible combinations, we develop an algorithm to efficiently discover only *maximal unions*. This problem is formally modeled as Maximal Biclique Enumeration (see Appendix A.7), which effectively identifies groups of reasoning subtrees and their shared subtree relations.

As illustrated in Figure 2(b), the reasoning trees for "Nobel Prize in Literature laureates" and "Booker Prize winners" both contain subtrees where second-layer entities (authors) are connected to third-layer entities via relations like 'has_nationality' and 'has_name'. Our method identifies this shared subtree structure. Relations not shared across all sets of subtrees, such as 'has_gender' (present only in the Nobel tree), are discarded during the union.

Once a maximal union is identified, we leverage an LLM to synthesize a question based on the common features of the selected subtrees. For instance, the question "Which authors have won both the Nobel Prize in Literature and the Booker Prize?" requires identifying the two sets of laureates as intermediate 'Target Entities' and then finding their intersection to produce the final 'Target Entities'. The complete walkthrough is in Appendix A.8.2.

### 3.1.3 VERSION-III: Reverse-Union

While the Union method generates complex, multi-source tasks, a vulnerability remains: an agent could solve the query and use direct keyword searches on the constituent sources (e.g., search 'Nobel Prize winners,' then 'Booker Prize winners'). This approach circumvents the intended synthesis of information, reducing the cognitive load and failing to stimulate true reasoning capabilities similar to WebSailor (Li et al., 2025c). To address this, we introduce Reverse-Union, a paradigm designed to enforce a more robust cognitive workflow by reversing the standard reasoning flow. As illustrated in Figure 2(c), this method combines two stages to construct a challenging task:

- **Deductive Fuzz:** This stage implements the fuzz by defining the 'Question Entities' as a set of descriptive third-layer entities. Instead of being named directly, a central 'anchor' entity (an entity from the second layer) is described through its corresponding third-layer entities. In the example, the description "the 1980s prize-winner that wrote a novel about a group of British boys stranded on an uninhabited island" serves as clues in the form of 'Question Entities'. An agent must first deduce from these clues to identify the anchor entity, 'William Golding'.

- **Union-based Search Construction:** After fuzzing the anchor, this stage constructs the expansive search part of the task, ensuring the anchor serves only as a bridge to the final answer. To achieve

this, we first select a specific third-layer entity from the anchor's subtree (e.g., his country) to act as a pivot. We then formulate the remainder of the question to compel an agent to use this pivot to launch a new search across the unified trees. The final Target Entities are thus defined as the set of second-layer entities that share this pivot attribute (i.e., are also British) and satisfy the original intersection condition (i.e., winning both prizes).

By structuring tasks this way, `Reverse-Union` prevents agents from succeeding with simple keyword searching and mandates a more robust, multi-step reasoning process. The detailed process of question generation and the required reasoning path are explained in Appendix A.8.3.

## 3.2 Information-Guided Trajectory Construction

After synthesizing the task, this section elaborates on the construction of task-solving trajectories. As shown in Eq.(3), our agent solves a task within the ReAct framework (Yao et al., 2023). We equip the agent with the following tools:

- `Search` This action enables the agent to conduct Google search by several queries. The parameters of this tool are $\{queries, filter\_year\}$, enabling temporal filtering of search results. This tool would return the top relevant URLs and their snippets as the observation.
- `Visit` This action enables the agent to visit multiple URLs. The parameters of this tool are $\{urls, goal\}$. This tool would return the summarized visited paragraphs as the observation.

After generating a large set of trajectories by executing our constructed tasks with an open-source model, we apply a filtering procedure to select high-quality examples for training. Our goal is to retain trajectories that demonstrate both accuracy in collecting the required entities and efficiency in the use of actions, in accordance with the metrics defined in Section 2.3. Specifically, we impose the following selection criteria:

**Coverage Criterion.** We require that the trajectory achieve sufficient completeness in information collection. Formally, we keep only those trajectories whose ISR satisfies $\text{ISR} > \alpha$, where $\alpha$ is a predefined coverage threshold. To compute ISR, we accumulate the obtained target entities in all actions. We compute ISR as Eq.( 4).

**Efficiency Criterion.** We further require that the trajectory maintain high efficiency in discovering useful entities. This translates into selecting those trajectories whose ISE satisfies $\text{ISE} > \beta$, where $\beta$ is a predefined efficiency threshold. For ISE, we accumulate the obtained target entities in `Visit` actions. The reason for not including `Search` in ISE is that we observe entities found in `Search` are less precise and would be updated by the following `Visit` action. We compute ISR as Eq.(5).

Through this filtering process, we ensure that the retained trajectories are both accurate in acquiring the target entities and efficient in their action usage, providing strong supervision signals for training agents to perform precise and effective information-seeking. This efficiency-oriented filtering is effective because it selects trajectories with high valid action density to teach focused planning and reduce redundancy, while our entity-intensive tasks provide sufficient high-quality trajectories and stabilize ISE ($\text{Var}(\text{ISE}) = \mathcal{O}\left(\frac{1}{n}\right)$) as a reliable signal, forming a mutually reinforcing cycle.

## 4 Experiments

### 4.1 Setup

**Benchmarks** We conduct extensive evaluations of our method on five challenging QA benchmarks that demand complex information-seeking capabilities, namely BrowseComp (Wei et al., 2025), GAIA (Mialon et al., 2023), xbench-DeepSearch (xbench-DS) (Xbench-Team, 2025), Seal-0 (Pham et al., 2025), and WideSearch (Wong et al., 2025). For GAIA, we adopt the 103-sample text-only validation subset (Li et al., 2025d), while for all other benchmarks, we utilize their complete test sets.

**Baselines** We select a representative set of mainstream and competitive information-seeking agents as our baselines, including proprietary agents (Claude-4-Sonnet (Anthropic, 2025), OpenAI-o3 (OpenAI, 2025a), OpenAI DeepResearch (OpenAI, 2025b)) and open-source agents (ASearcher (Gao et al., 2025), DeepDive (Lu et al., 2025), DeepDiver-V2 (Team), MiroThinker (Team et al., 2025b), Kimi-K2 (Team et al., 2025a), WebExplorer (Liu et al., 2025), WebDancer (Wu et al., 2025a), WebSailor (Li et al., 2025c), WebShaper (Tao et al., 2025)).

Table 1: Main results on multiple benchmarks. All benchmarks except WideSearch report `Pass@1`. WideSearch reports Success Rate (SR), `Row F1`, and `Item F1`. **Bold** scores indicate the highest values among all open-source agents.

| Model / Framework | BrowseComp | GAIA | xbench-DS | Seal-0 | WideSearch | | |
|---|---|---|---|---|---|---|---|
| | | | | | SR | Row F1 | Item F1 |
| *Proprietary Agents* | | | | | | | |
| Claude-4-Sonnet | 12.2 | 68.3 | 64.6 | – | 2.3 | 31.7 | 57.9 |
| OpenAI-o3 | 49.7 | 70.5 | 66.7 | 18.9 | 4.5 | 34.0 | 52.6 |
| OpenAI DeepResearch | 51.5 | 67.4 | – | – | – | – | – |
| *Open-Source Agents* | | | | | | | |
| ASearcher-Web-32B | 5.2 | 52.8 | 42.1 | – | – | – | – |
| DeepDive-32B | 14.8 | – | 50.5 | – | – | – | – |
| DeepDiver-V2-38B | 13.4 | – | 53.0 | – | – | – | – |
| MiroThinker-32B-DPO-v0.2 | 13.0 | 64.1 | – | – | – | – | – |
| Kimi-K2-Instruct-1T | 14.1 | 57.7 | 50.0 | – | 1.1 | **29.7** | **54.4** |
| WebExplorer-8B | 15.7 | 50.0 | 53.7 | – | – | – | – |
| WebDancer-QwQ-32B | 3.8 | 51.5 | 38.3 | – | 0.0 | 9.3 | 34.5 |
| WebSailor-32B | 10.5 | 53.2 | 53.3 | 21.3 | 0.0 | 2.1 | 5.5 |
| WebShaper-QwQ-32B | – | 53.3 | 35.0 | – | 0.0 | 9.9 | 31.5 |
| WebLeaper-Union | 22.1 | **69.9** | 62.3 | 35.1 | **4.0** | 22.2 | 34.5 |
| WebLeaper-Reverse-Union | **23.0** | 67.0 | **66.0** | **37.2** | **4.0** | 25.8 | 40.8 |

**Training Configurations** To maintain the basic deep search ability, we combine our data with 5,000 `WebSailor-V2` (Li et al., 2025b) data to train the model. We separately merge 5,000 `WebSailor-V2` data with `Basic`, `Union`, and `Reverse-Union` data of `WebLeaper`, which stimulates the IS ability to a larger degree (with $\alpha$ in ISR set to 0.3 and $\beta$ in ISE set to 0.1). On this basis, we employ `Qwen3-30B-A3B-Thinking-2507`[1] as the base model, trained using `Megatron` framework[2]. The training follows a standard SFT procedure. In our experiments, SFT on the final dataset (approximately 15k samples) was completed in about 6 to 8 hours on a cluster of 64 H20 GPUs.

**Evaluation Metrics and Inference Hyper-parameters** The overall evaluation follows the settings specified by each benchmark. For BrowseComp, GAIA, xbench-DS, and Seal-0, we report the `pass@1` scores obtained via LLM-as-a-judge evaluation as the final results. For WideSearch, we report the success rate (SR) for fully retrieving all target results, along with two F1 scores—`Row F1` and `Item F1`—which are computed using a combination of string matching and LLM-as-a-judge evaluation, in alignment with the official evaluation protocol. During LLM inference, we configure the sampling parameters (temperature and top-$p$) to 0.6 and 0.95, respectively.

## 4.2 OVERALL PERFORMANCE

As shown in Table 1, `WebLeaper` achieves state-of-the-art performance compared to mainstream open-source agents on five challenging information-seeking QA benchmarks. Notably, on benchmarks other than BrowseComp and WideSearch, it even delivers performance comparable to, or surpassing, that of agents built on `Claude-4-Sonnet` and `OpenAI-o3`. Even on the highly challenging BrowseComp benchmark, `WebLeaper` significantly outperforms `Kimi-K2-Instruct-1T`, despite the latter having a much larger parameter scale. It is also worth noting that the `Reverse-Union` data, which incorporates greater task complexity on top of the `Union` data, employs an fuzz strategy that further facilitates the model's ability to integrate information-seeking with planning and reasoning, thereby enhancing its overall information-seeking QA capability.

Crucially, `WebLeaper`'s robustness is demonstrated by its strong performance on test sets that are not limited to Wikipedia. While our agent is trained on Wikipedia-derived data, it is evaluated on five diverse, real-world benchmarks that require querying and reasoning over the entire live web. The consistent and significant improvements across these varied benchmarks strongly suggest that our method learns a generalizable skill of efficient information seeking, rather than overfitting to the specific structure of Wikipedia.

---

[1]`https://huggingface.co/Qwen/Qwen3-30B-A3B-Thinking-2507`
[2]`https://github.com/NVIDIA/Megatron-LM`

Table 2: Ablation study on training results across different data sources (for efficiency considerations, we use the `WideSearch` (English subset) and `BrowseComp` (200 subset), while the full sets are used for the other benchmarks). Numbers in parentheses denote the difference compared to training only with the `WebSailor-V2-5k` data. † denotes a mixed version that includes the `WebSailor-V2-5k` data.

| Data Source | BrowseComp | WideSearch | GAIA | Seal-0 | xbench-DS | Avg. |
|---|---|---|---|---|---|---|
| `WebSailor-V2-5k` | 25.17 | 33.15 | 67.69 | 34.23 | 60.00 | 44.05 |
| `WebSailor-V2-10k` | 24.50 | 38.91 | 66.02 | 33.93 | 62.67 | 45.21 |
| `Basic-5k`† | 20.67 (-4.50) | 32.26 (-0.89) | 40.78 (-26.91) | 30.03 (-4.20) | 58.33 (-1.67) | 36.41 (-7.64) |
| `Union-5k`† | 27.50 (+2.33) | 41.70 (+8.55) | 69.90 (+2.21) | 35.14 (+0.82) | 62.33 (+2.33) | 47.31 (+3.26) |
| `Reverse-Union-10k`† | 27.67 (+2.50) | 44.07 (+10.92) | 66.99 (-0.70) | 37.24 (+3.01) | 66.00 (+6.00) | 48.39 (+4.34) |

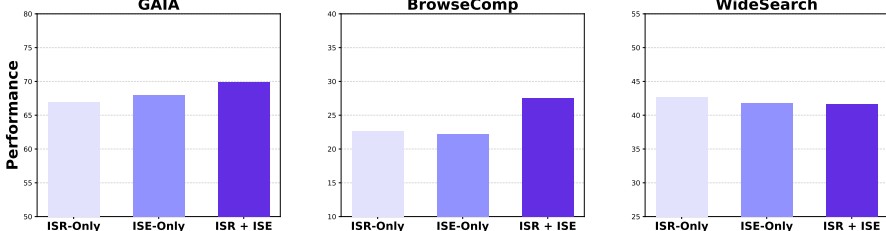

Figure 3: Ablation study results on information-guided trajectory construction strategies.

Overall, the observed performance improvements validate that our proposed approaches—entity-intensive task synthesis and information-guided trajectory construction—significantly enhance the agent's information-seeking capabilities, even under a modest parameter budget.

### 4.3 CAPABILITY GAINS INDUCED BY ENTITY-INTENSIVE TASK SYNTHESIS

To investigate the effectiveness of our entity-intensive task synthesis method, we conduct a comparative analysis against training solely on the `WebSailor-V2` dataset (using 5,000 and 1,000 samples, respectively), a synthetic corpus specifically designed to stimulate the agent's deep search capability.

As shown in Table 2, we investigate the impact of different entity-intensive task synthesis strategies through an ablation study on all these benchmarks. The `Basic` setting exhibits substantial drops across all three datasets compared to `WebSailor-V2-5k`. This poor performance can be attributed to the inherent limitations of the Basic data construction method: tasks generated under this setting tend to be overly simple, allowing the model to infer complete answers from only a few information sources. Such shortcut patterns encourage the model to overfit to superficial cues rather than learning to integrate diverse information, ultimately impairing generalization.

In contrast, the `Union` strategy consistently outperforms `WebSailor-V2-5k`, achieving an average improvement of +3.26. By combining heterogeneous information sources and increasing the complexity of task construction, `Union` mitigates the shortcut problem inherent in `Basic`, forcing the model to reason over dispersed and complementary evidence. This leads to more robust performance across datasets and demonstrates the effectiveness of the proposed data construction approach.

Furthermore, compared to `Union`, `Reverse-Union` introduces a certain degree of reasoning complexity into the information-seeking process, making it more challenging for the model to readily identify where to begin entity retrieval. This design particularly enhances the model's planning and decision-making capabilities in information-seeking tasks. The improvement in these abilities is clearly reflected in performance, leading to substantial and widespread gains across all benchmarks.

### 4.4 IMPACT OF INFORMATION-GUIDED TRAJECTORY CONSTRUCTION

We compare the proposed information-guided trajectory construction strategies across `ISR-Only`, `ISE-Only`, and `ISR+ISE` on three representative benchmarks—GAIA, BrowseComp, and WideSearch—to examine the independent and combined effects of ISE and ISR.

On GAIA and BrowseComp, `ISR+ISE` achieves the best performance, suggesting that integrating precision and efficiency constraints produces trajectories that are both goal-directed and concise, thereby reducing redundant exploration. This indicates that in more complex browsing tasks, relevance and efficiency constraints complement each other to generate higher-quality trajectories.

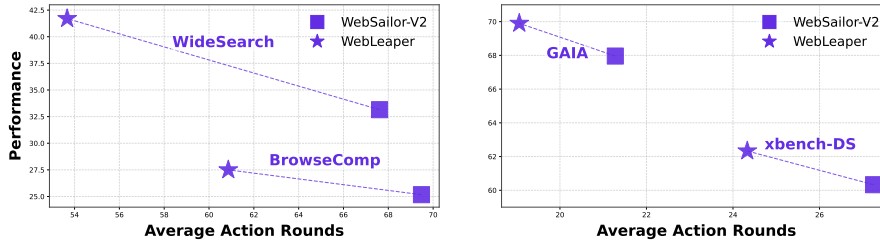

Figure 4: Effectiveness and efficiency comparison between `WebLeaper` and `WebSailor-V2`.

In contrast, on WideSearch, the three strategies deliver comparable results, with performance differences falling within the margin of variance. This suggests that for broad search tasks, the specific choice of trajectory filtering plays a less critical role—likely because training on entity-intensive synthesized data already provides strong broad search capabilities.

## 4.5 JOINT GAINS IN EFFICIENCY AND EFFECTIVENESS

As illustrated in Figure 4, `WebLeaper` consistently outperforms the baseline in terms of both effectiveness and efficiency. In the WideSearch and BrowseComp benchmarks, our approach achieves markedly higher performance scores while requiring fewer average action rounds, indicating that the search process is not only more accurate but also more efficient. Similarly, in the GAIA and xbench-DS tasks, our method improves effectiveness while simultaneously reducing the operational cost. This demonstrates that our design enables a more targeted search strategy, resulting in reduced interaction steps without sacrificing—and in fact enhancing—the quality of the results.

Overall, these results validate that our proposed method achieves superior joint optimization of information-seeking efficiency and task performance compared to the baseline. This reflects our key insight: an agent should not merely learn to search, but rather learn to search efficiently and wisely, thereby achieving a better balance between efficiency and effectiveness.

## 4.6 EXPLORATION ON HYPER-PARAMETERS ISR $\alpha$ AND ISE $\beta$

To validate the robustness of our information-guided trajectory construction (Section 3.2), we conduct a sensitivity analysis of the core filtering hyperparameters: $\alpha$ (Information-Seeking Rate, ISR, threshold) and $\beta$ (Information-Seeking Efficiency, ISE, threshold) (both defined in Section 2.3). $\alpha$ ensures trajectory accuracy by retaining only sequences with sufficient target entity coverage, while $\beta$ enforces efficiency by excluding overly time-consuming trajectories.

Table 3: Performance (Pass@1) on Browser-Comp (200 subset) for different $\alpha$ (ISR threshold) and $\beta$ (ISE threshold) combinations. A value of 0 disables the filter.

| $\beta \setminus \alpha$ | 0.3 | 0 |
|---|---|---|
| 0.1 | **27.5** | 22.2 |
| 0 | 14.0 | 13.7 |

We trained `WebLeaper-Reverse-Union` with different $\alpha/\beta$ combinations, using BrowserComp (200 subset) for evaluation; a threshold of 0 disables the corresponding filter. Table 3 shows the results: the highest performance is achieved with $\alpha = 0.3$ and $\beta = 0.1$ (our current configuration). Disabling either filter degrades performance, and no filtering yields the lowest score, confirming both constraints are critical for high-quality training data.

The selection of $\alpha = 0.3$ and $\beta = 0.1$ balances data quality and quantity. Excessively high thresholds would shrink the dataset below the sample size of baselines, while overly low thresholds introduce low-quality trajectories. This "sweet spot" preserves a comparable sample size while removing poor-quality sequences, ensuring fair baseline comparisons and validating our parameter choice.

## 4.7 RESULTS ON DIFFERENT BACKBONE

To showcase its broader applicability across diverse model architectures, we evaluate our method on multiple backbones, placing particular emphasis on the `Qwen3-4B-Thinking-2507` model. For this comprehensive evaluation, we adopt two training configurations: the baseline setup entails exclusive fine-tuning on 5,000 high-quality samples from the `WebSailor-V2` dataset, while our proposed

Table 4: SFT results on `Qwen3-4B-Thinking-2507`. **Bold** scores indicate the highest values.

| Model / Framework | BrowseComp | GAIA | xbench-DS | Seal-0 | WideSearch | | |
|---|---|---|---|---|---|---|---|
| | | | | | SR | Row F1 | Item F1 |
| *Qwen3-4B-Thinking-2507* | | | | | | | |
| WebSailor-V2 | 16.2 | 55.3 | 53.3 | 30.9 | 1.5 | 15.7 | 30.3 |
| WebLeaper-Reverse-Union | **17.7** | **58.3** | **54.7** | **32.1** | **3.5** | **21.1** | **38.1** |

method leverages a mixed training corpus that integrates 5,000 `WebSailor-V2` samples with our carefully curated `WebLeaper-Reverse-Union` data.

As illustrated in Table 4, even when deployed on the relatively compact `Qwen3-4B-Thinking-2507`, training augmented with `WebLeaper` data delivers consistent and substantial performance enhancements across all evaluated benchmarks relative to the baseline. This result underscores the fundamental nature of the advantages brought by entity-intensive task synthesis and information-guided trajectory construction strategies—advantages that transcend constraints imposed by specific model scales or architectural paradigms, thus validating the generalizability of our approach.

## 5 RELATED WORK

### 5.1 INFORMATION SEEKING AGENT

LLM-powered information-seeking agents can be broadly categorized into 3 streams: (1) enhancing core models via supervised fine-tuning (Zeng et al., 2023; Wu et al., 2025a; Li et al., 2025c;b; Tao et al., 2025; Su et al., 2025; Fang et al., 2025a); (2) advancing agent architecture for improved planning and robustness (Qiao et al., 2025; Xu et al., 2025a; Li et al., 2025a); and (3) developing multi-agent systems for collaborative problem-solving (Wu et al., 2023; Hong et al., 2024). Our work aligns with the first category but addresses a key limitation. Prior methods often train on tasks focused on correctness with single-fact answers, which is insufficient for large-scale information gathering. We posit that the number of entities in an answer—its entity richness—is a critical dimension for evaluating an agent's completeness and efficiency. This paper aims to bridge this gap by creating and utilizing entity-rich QA data to enhance agent capabilities for comprehensive information acquisition.

### 5.2 AGENT DATA SYNTHESIS

Synthetic data generation is pivotal for agent training, with primary applications in tool use (Wu et al., 2025a; Tao et al., 2025; Shen et al., 2025; Fang et al., 2025b), code generation (Jimenez et al., 2024; SHEN et al., 2025; Xu et al., 2025c; Shao et al., 2025), and GUI automation (Xu et al., 2025b; Sun et al., 2025; Pahuja et al., 2025). These efforts primarily combat data scarcity. Within the information-seeking domain, existing data synthesis approaches increase task difficulty through multi-step reasoning (Wu et al., 2025b;a; Tao et al., 2025) or long-horizon planning (Qiao et al., 2025). We contend that such methods often overlook the semantic richness of the training data itself. In contrast, our approach centers on synthesizing QA data with high entity-level complexity. We hypothesize that this focus on data semantics is a crucial and complementary path to improving agent reasoning and world knowledge alignment.

## 6 CONCLUSION

In this paper, we addressed low search efficiency in LLM-based information-seeking agents. We argued that the sparsity of target entities in conventional training tasks is a primary contributor to this inefficiency. To overcome this, we introduced `WebLeaper`, a framework for constructing entity-intensive IS tasks and generating efficient solution trajectories. By formulating IS as a tree-structured reasoning problem and scaling task complexity through `Basic`, `Union`, and `Reverse-Union` synthesis, we built a rich training environment. Our ISR/ISE-guided trajectory curation further ensures training on solutions that are both accurate and efficient. Extensive experiments on five benchmarks show that `WebLeaper`consistently improves performance, confirming that improving search efficiency is a strong lever for enhancing overall IS-agent capability.

## ACKNOWLEDGEMENTS

This work is supported by National Natural Science Foundation of China (92470121, 62402016, 62595734), National Key R&D Program of China (2024YFA1014003), Zhongguancun Academy (Grant No.s C20250204, C20250602), and High-performance Computing Platform of Peking University.

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

# A  APPENDIX

## A.1  DECLARATION ON THE USE OF LLMS

We declare that the use of LLMs during the preparation of this manuscript was strictly limited to language-related assistance, such as sentence refinement and grammatical correction. All substantive content was independently authored by the authors and rigorously reviewed and verified following any LLM-assisted modifications. During the experiments, all usage of LLMs was solely for academic research purposes, with no inappropriate applications. Detailed experimental settings are provided in the Experiments section of this paper. No other reliance on LLMs is involved in this work.

## A.2  PROOF OF PROPOSITION 1

This appendix provides the detailed mathematical derivation for Proposition 1, as presented in Section 2.3. The purpose of this proof is to formally establish that the variance of the Information-Seeking Efficiency (ISE) metric is inversely proportional to $n$, the number of required entities. This property, $\mathrm{Var}(\mathrm{ISE}) = \mathcal{O}(1/n)$, demonstrates that ISE becomes an increasingly stable and reliable performance measure as the complexity of the task (i.e., the size of $n$) grows.

*Proof.* Recall from the main text that the Information-Seeking Efficiency (ISE) is defined as
$$\mathrm{ISE} = \frac{n}{T}, \tag{7}$$
where $T$ is the total number of action steps taken to discover all $n$ required distinct entities in the set $R$.

**Step 1: Modelling the random variables.** Let $X_i$ denote the number of steps the agent spends to discover the $i$-th new required entity in $R$. By assumption:

- $X_1, \ldots, X_n$ are independent and identically distributed.

- $\mu := \mathbb{E}[X_i] > 0$ (finite mean).

- $\sigma^2 := \mathrm{Var}(X_i) < \infty$ (finite variance).

- $X_i > 0$ almost surely (each discovery takes a positive number of steps).

The total number of steps is
$$T = \sum_{i=1}^{n} X_i.$$

**Step 2: Sample mean and basic properties.** Define the sample mean
$$\overline{X} := \frac{T}{n} = \frac{1}{n} \sum_{i=1}^{n} X_i.$$
By standard properties of i.i.d. random variables:
$$\mathbb{E}[\overline{X}] = \mu, \quad \mathrm{Var}(\overline{X}) = \frac{\sigma^2}{n}. \tag{8}$$
From the ISE definition Eq.(5), we have the exact identity
$$\mathrm{ISE} = \frac{1}{\overline{X}}.$$

**Step 3: Reduction to the variance of a smooth function.** We have $\mathrm{ISE} = f(\overline{X})$ with $f(x) = x^{-1}$. Since $\mu > 0$ and $X_i > 0$ a.s., the function $f$ is infinitely differentiable in a neighborhood of $\mu$.

Write
$$\delta_n := \overline{X} - \mu$$
so that $\mathbb{E}[\delta_n] = 0$ and $\mathrm{Var}(\delta_n) = \sigma^2/n$. By the Strong Law of Large Numbers, $\delta_n \to 0$ almost surely as $n \to \infty$. Thus, for large $n$, with high probability $|\delta_n| < \mu/2$, ensuring that a Taylor expansion of $f$ around $\mu$ is valid.

**Step 4: Taylor expansion with remainder control.** On the event $\{|\delta_n| < \mu/2\}$, we expand $f(\mu + \delta_n)$ using Taylor's theorem to second order:

$$\frac{1}{\mu + \delta_n} = \frac{1}{\mu} - \frac{\delta_n}{\mu^2} + \frac{\delta_n^2}{\mu^3} + R_3(\delta_n), \tag{9}$$

where the remainder term satisfies $|R_3(\delta_n)| \leq C|\delta_n|^3$ for some $C > 0$ depending only on $\mu$.

**Step 5: Mean and second moment calculations.** From Eq.(9):

$$\mathbb{E}\left[\frac{1}{\overline{X}}\right] = \frac{1}{\mu} + \frac{\mathbb{E}[\delta_n^2]}{\mu^3} + \mathbb{E}[R_3(\delta_n)], \tag{10}$$

$$\mathbb{E}\left[\frac{1}{\overline{X}^2}\right] = \frac{1}{\mu^2} + \frac{\mathbb{E}[\delta_n^2]}{\mu^4} + O\big(\mathbb{E}[|\delta_n|^3]\big). \tag{11}$$

By Eq.(8), $\mathbb{E}[\delta_n^2] = \sigma^2/n$. Also, finite variance plus Hölder's inequality yields $\mathbb{E}[|\delta_n|^3] = O(n^{-3/2})$. Therefore:

$$\mathbb{E}[1/\overline{X}] = \frac{1}{\mu} + \frac{\sigma^2}{\mu^3 n} + o\left(\frac{1}{n}\right), \tag{12}$$

$$\mathbb{E}[1/\overline{X}^2] = \frac{1}{\mu^2} + \frac{\sigma^2}{\mu^4 n} + o\left(\frac{1}{n}\right). \tag{13}$$

**Step 6: Computing the variance.** Using $\mathrm{Var}(Y) = \mathbb{E}[Y^2] - (\mathbb{E}[Y])^2$ with $Y = 1/\overline{X}$:

$$\mathrm{Var}(\mathrm{ISE}) = \left(\frac{1}{\mu^2} + \frac{\sigma^2}{\mu^4 n} + o\left(\frac{1}{n}\right)\right) - \left(\frac{1}{\mu} + \frac{\sigma^2}{\mu^3 n} + o\left(\frac{1}{n}\right)\right)^2$$

$$= \frac{\sigma^2}{\mu^4 n} + o\left(\frac{1}{n}\right),$$

where the constant term $\frac{1}{\mu^2}$ cancels exactly.

**Step 7: Conclusion.** We have shown that

$$\mathrm{Var}(\mathrm{ISE}) = \frac{\sigma^2}{\mu^4 n} + o\left(\frac{1}{n}\right),$$

which in particular implies the order bound $\mathrm{Var}(\mathrm{ISE}) = \mathcal{O}(n^{-1})$.

This completes the proof. $\square$

### A.3 Reinforcement Learning Experiments

To further explore the potential of `WebLeaper`'s data, we conduct reinforcement learning (RL) experiments by applying GRPO to our advanced SFT model. The core goal is to verify whether integrating RL can further boost the performance of our best-performing SFT model, leveraging RL's strength in optimizing action quality.

RL has proven effective in advanced reasoning tasks by exploring logical step spaces, but its objective in the information-seeking domain is distinct: optimizing the quality of external tool actions (e.g., search, visit). We design a dense, granular reward signal based on a soft F-score that combines recall (ISR) and precision, creating a rich and accurate learning signal. This reward mainly encourages the agent to prioritize "better steps", which is our first motivation. The length of the reasoning content main also varies properly for this purpose.

Table 5 presents the performance comparison between the SFT baseline and SFT+RL. Results show that RL significantly improves performance across all benchmarks: GAIA sees a 3.3-point gain, xbench-DS improves by 3.0 points, and WideSearch's Success Rate (SR) jumps by 2.5 points. This validates that RL, when combined with `WebLeaper`'s high-quality synthetic data, effectively enhances the agent's action optimization capability in information-seeking tasks.

### A.4 Clarification on Tool Call Efficiency and Reasoning Content Length

Our core objective in introducing ISE filtering is to enhance the *quality and efficiency of tool calls*, not to restrict the length or depth of reasoning content. These two dimensions are fundamentally

| Model | BrowseComp | GAIA | xbench-DS | WideSearch (SR) | WideSearch (Item F1) |
|---|---|---|---|---|---|
| SFT | 37.8 | 69.9 | 69.0 | 1.5 | 45.4 |
| SFT+RL | 38.8 (+1.0) | 73.2 (+3.3) | 72.0 (+3.0) | 4.0 (+2.5) | 48.5 (+3.1) |

Table 5: Performance comparison between SFT and SFT+RL across benchmarks.

distinct: tool call efficiency focuses on minimizing redundant, irrelevant, or unnecessary interactions with external tools (e.g., redundant searches, unfocused URL visits), while reasoning content length refers to the performance of the agent's internal cognitive process (e.g., step-by-step chain-of-thought (CoT) reasoning, intermediate deduction for complex constraints). The ISE metric targets the former—ensuring each tool call contributes meaningfully to entity retrieval—without imposing arbitrary limits on the latter.

Notably, the length of reasoning content dynamically adapts to serve our efficiency goal and the inherent complexity of the task. For tasks requiring deep exploration (e.g., multi-step logical deduction, adaptive constraint satisfaction), the agent's reasoning process naturally expands to accommodate structured planning (e.g., decomposing complex queries into subgoals, verifying intermediate entities). However, this expanded reasoning is directed toward optimizing tool call precision—e.g., generating more targeted search queries or validating the relevance of retrieved information—rather than indulging in unfocused divergent exploration. For complex tasks like AIME-style problems or long-form CoT reasoning, our framework preserves the necessary cognitive depth by allowing reasoning content to scale with task difficulty, while ISE filtering only suppresses *unproductive tool calls* (e.g., repetitive searches with minor keyword variations) that do not advance the reasoning process.

In summary, strict ISE constraints do not hinder adaptive exploration or complex reasoning. Instead, they decouple tool call efficiency from reasoning length: reasoning content evolves to support targeted, high-quality tool interactions, and task complexity inherently shapes the depth of reasoning—all while ISE ensures tool usage remains purposeful and resource-efficient. This design balances exploration and efficiency, avoiding both tool call redundancy and overly rigid reasoning suppression.

### A.5 DATA STATISTICS

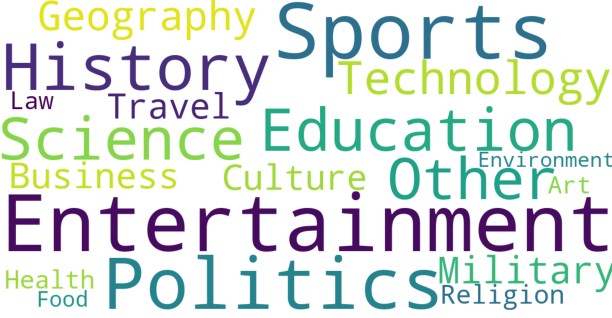

Figure 5: The distribution of our training data.

Figure 5 illustrates the distribution of our training data.

Figure 6 displays the entity count distribution of our training data. A significant portion of our samples contain at least 100 entities, underscoring the inherent difficulty of our dataset. As formalized in Equation 6, this complexity is crucial for robustly measuring efficiency, which in turn leads to improved overall performance.

### A.6 DATA CLEANING AND BASIC TASK CONSTRUCTION

This section elaborates on the data processing and construction methodology for the `Basic` version tasks introduced in Section 3.1.1.

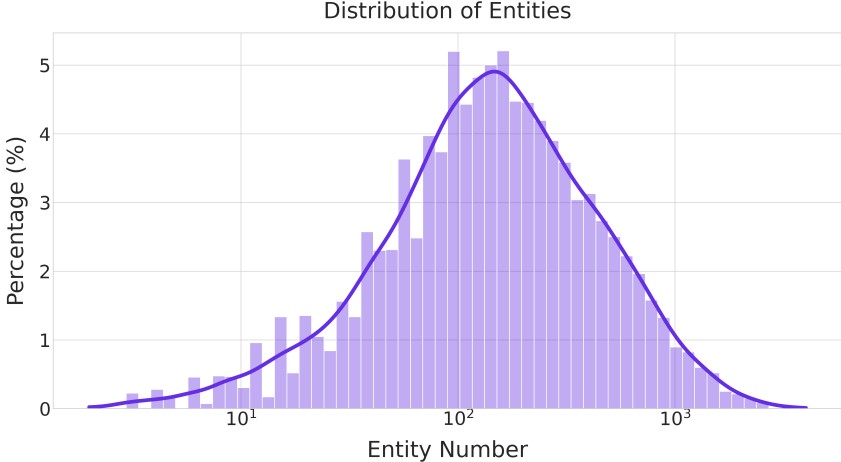

Figure 6: Entity Count Distribution in Training Data. A significant portion of our samples contains at least 100 entities, underscoring the inherent difficulty of our dataset. This complexity, as formalized in Equation equation 6, is crucial for robustly measuring efficiency, which in turn contributes to improved overall performance.

**Rationale for Tree Structure**    In information-seeking tasks, the reasoning structure is paramount. We chose a tree structure for our basic tasks because it offers a compact and hierarchical organization of entities. This structure is highly efficient for representing a large number of interconnected entities that stem from a common query concept, mirroring many real-world information-gathering scenarios. A reasoning tree is composed of a root (question entity) and a set of subtrees, where each subtree represents a cohesive unit of information.

**Multi-Stage Table Cleaning**    To ensure the quality and suitability of the data used for task synthesis, we crawled approximately 2 million tables from Wikipedia and subjected them to a rigorous multi-stage cleaning procedure. This was essential because raw web tables are often noisy and inconsistent. The stages were as follows:

- **Size Filtering:** We first discarded tables that were either too small (fewer than 10 rows or 3 columns) to capture meaningful relational information, or too large (more than 200 rows or 20 columns) to be processed efficiently and form a coherent task.
- **Semantic and Structural Filtering:** We then removed semantically irrelevant columns that frequently appear in web tables, such as those containing serial numbers, notes, or references. Tables with significant formatting errors (e.g., numerous merged cells that disrupt the relational structure) were also excluded.
- **Isomorphism and Homogeneity:** Finally, we retained only groups of isomorphic tables (tables sharing the same column headers and structure). This step was crucial for ensuring structural homogeneity across our dataset, which is a prerequisite for identifying common subtree structures needed for the `Union` operation described later.

The resulting collection contains clean, well-structured tables with a set of meaningful fields as columns and multiple rows, where each row can be transformed into a subtree.

**Reasoning Tree Population**    To construct the three-layer reasoning tree from a single table, we populate the layers as follows:

- **First Layer (Question Entities):** Entities mentioned in the table's title or caption are extracted to form the root of the tree.
- **Second Layer (Roots of Subtrees):** We employ an LLM to analyze the table's columns and select one that contains no duplicate entries. This column is treated as the key, and its values become the second-layer entities of the tree. Each of these entities serves as the root of a subtree. The LLM is effective at identifying columns like 'Name' or 'Title' that serve this unique identification purpose.

- **Third Layer (Leaves of Subtrees):** The values in the remaining columns of the table constitute the third layer, representing the leaf entities associated with each second-layer entity.

## A.7 MAXIMAL UNION ALGORITHM FOR TASK SYNTHESIS

This section provides the formal definition and algorithmic implementation for discovering maximal union groups, as introduced in Section 3.1.2. The core of our approach is to reformulate the search for compatible reasoning trees as a Maximal Biclique Enumeration (refer to 1 problem on a bipartite graph.

**Problem Formulation**

Let $\mathcal{T}_{\text{base}} = \{T_1, T_2, \ldots, T_N\}$ be our collection of basic reasoning trees. We first construct a bipartite graph $G = (U, V, E)$, where $U = \mathcal{T}_{\text{base}}$ is the set of all trees, and $V$ is the set of all unique relation names found within the subtrees across all trees in $\mathcal{T}_{\text{base}}$. An edge $(T_i, v_j) \in E$ exists if the relation $v_j$ is present in any subtree of tree $T_i$ (i.e., $v_j \in \text{Rel}(T_i)$, where $\text{Rel}(T_i) = \bigcup_k \text{Rel}(S_{i,k})$).

In this construction, a *maximal union* directly corresponds to a *maximal biclique* $(\mathcal{U}, \mathcal{V})$, where $\mathcal{U} \subseteq U$ is a set of trees and $\mathcal{V} \subseteq V$ is a set of their common relations. Our goal is to find all such maximal bicliques that satisfy certain size and semantic constraints. Formally, we seek to find all maximal pairs $(\mathcal{U}, \mathcal{V})$ that satisfy:

$$\begin{aligned} \text{find maximal} \quad & (\mathcal{U}, \mathcal{V}) \\ \text{subject to} \quad & \forall T_i \in \mathcal{U}, \mathcal{V} \subseteq \text{Rel}(T_i), \\ & |\mathcal{U}| \geq k_{\min}, |\mathcal{V}| \geq m_{\min}. \end{aligned} \quad (14)$$

Here, maximality means that no other tree can be added to $\mathcal{U}$ and no other relation can be added to $\mathcal{V}$ without violating the biclique property. Solving this by reformulating it as a standard maximal biclique enumeration problem is computationally efficient compared to an exhaustive search.

**Algorithm and Implementation Details**

- **Input:** A collection of base reasoning trees $\mathcal{T}_{\text{base}}$; a minimum number of trees for a valid union, $k_{\min}$; a minimum number of common relations, $m_{\min}$.

- **Goal:** To find all *maximal union groups*, which are the solutions $(\mathcal{U}, \mathcal{V})$ to Eq. (14) that also satisfy the semantic matching criteria below.

- **Subtree Relation Matching Criteria:** To ensure the semantic coherence of unions, we impose strict matching criteria. For relations connecting the second and third layers, we require they share the same standardized name, data type, and domain. For the second-layer entities themselves (the roots of the subtrees), we relax this constraint, requiring only a match in data type and domain. This flexibility allows for the union of trees with conceptually similar but differently named second-layer entities (e.g., fusing a tree where entities are 'Authors' with another where they are 'Writers').

- **Output:** A set of maximal union groups $\mathcal{F}$, where each element is a tuple $\langle U', V' \rangle$ that meets the specified criteria.

The process is detailed in Algorithm 1.

## A.8 DETAILED EXAMPLES OF TASK SYNTHESIS

This section provides detailed explanations and reasoning walkthroughs for the examples of the three task synthesis versions presented in Section 3 and Figure 2.

### A.8.1 VERSION-I: BASIC

The goal of the basic version is to create a task with a clear, hierarchical reasoning structure derived from a single, self-contained set of entities.

**Example Question:** *Who were the Nobel Prize winners in Literature between 1980 and 1990? Please include their name, country, award year, and gender.*

---

**Algorithm 1:** Maximal Union Identification Algorithm

---

**Input:** A collection of base reasoning trees $\mathcal{T}_{\text{base}}$, minimum trees $k_{\min}$, minimum common relations $m_{\min}$.

**Output:** A set of maximal union groups $\mathcal{F}$.

1 $\mathcal{F} \leftarrow \emptyset$;
   // 1. Construct the bipartite graph from trees and subtree relations
2 Let $U$ be the set of trees from $\mathcal{T}_{\text{base}}$ and $V$ be the set of unique standardized relation names found within the subtrees of all trees in $\mathcal{T}_{\text{base}}$;
3 Construct the graph $G = (U, V, E)$ where an edge $(u, v) \in E$ exists if tree $u$ contains the relation $v$ in its subtrees (i.e., $v \in \text{Rel}(u)$);
   // 2. Enumerate maximal bicliques from the graph
4 $\mathcal{B} \leftarrow \text{EnumerateMaximalBicliques}(G)$;
                                  // Leverages standard algorithms like MICA or Eclat
   // 3. Filter and validate bicliques to form final union groups
5 **for** *each maximal biclique* $(U', V')$ *in* $\mathcal{B}$ **do**
     // Check size constraints from Eq. (1)
6     **if** $|U'| < k_{\min}$ *or* $|V'| < m_{\min}$ **then**
7         **continue**;
     // Validate semantic compatibility of second-layer entities
8     Let $T_{\text{id}}, D_{\text{id}}$ be the type and domain of the second-layer entities of the first tree in $U'$;
9     is_compatible $\leftarrow$ **true**;
10    **for** *each tree* $u \in U'$ **do**
11       **if** *u's second-layer entity type* $\neq T_{id}$ *or domain* $\neq D_{id}$ **then**
12          is_compatible $\leftarrow$ **false**;
13          **break**;
     // If all checks pass, add to the set of valid union groups
14    **if** *is_compatible* **then**
15       $\mathcal{F} \leftarrow \mathcal{F} \cup \{\langle U', V' \rangle\}$;

16 **return** $\mathcal{F}$;

---

**Construction Process:** The task is constructed from a single Wikipedia table, forming a reasoning tree. The layers shown in Figure 2(a) are populated as follows:

- **First Layer (question entities):** Derived from the table's title and a specified constraint, forming the query's scope: *Literature Nobel Prize, year 1980–1990*.
- **Second Layer (subtree roots):** Populated from the table's key column (e.g., author names): *Czesław Miłosz, William Golding, . . . .*
- **Third Layer (subtree leaves):** Consists of values from the remaining columns, representing attributes for each second-layer entity. For example: *man, Poland, 1980* for Czesław Miłosz. The edges connecting the second to the third layer represent relations like 'has_gender', 'has_country', 'has_award_year'.

**Reasoning Path:** An agent is expected to follow this hierarchical structure:

- **Identify Scope:** Recognize the 'Question Entities' from the query: *Nobel Prize in Literature, 1980–1990*.
- **Retrieve Second-Layer Entities:** Retrieve the second-layer entities, which are the authors: Czesław Miłosz, William Golding, ....
- **Gather Attributes:** For each second-layer entity, follow the relations to retrieve their associated third-layer entities, such as Poland, 1980, man for Czesław Miłosz.

### A.8.2 VERSION-II: UNION

This version increases structural complexity by requiring the agent to perform relational operations across distinct reasoning trees.

**Example Question:** *Which authors have won both the Nobel Prize in Literature and the Booker Prize? For each, provide their name, nationality and the year they won the Nobel.*

**Construction Process:** Once a maximal union is identified (e.g., between the reasoning trees for 'Nobel Prize laureates' and 'Booker Prize winners,' which share common relations like 'has_nationality' within their subtrees), an LLM generates a task requiring information integration. The LLM is prompted to find an interesting relationship, such as the intersection of the two sets of second-layer entities (authors), and then weave this logic into a natural language question.

**Reasoning Path:** The task is constructed from a *maximal union* of two distinct reasoning trees. To solve this, an agent must:

- **Retrieve First Entity Set:** Identify the first concept, 'Nobel Prize in Literature,' and retrieve the full set of corresponding second-layer entities from the first tree, $R_{\text{Nobel (T1)}}$.

- **Retrieve Second Entity Set:** Identify the second concept, 'Booker Prize,' and retrieve its full set of second-layer entities from the second tree, $R_{\text{Booker (T2)}}$.

- **Find Intersection:** Perform a relational join to find the intersection of the two sets of second-layer entities based on name. The final 'Target Entities' are the entities present in both sets, such as {*William Golding*, *J.M. Coetzee*, ... }, along with their requested third-layer attributes.

### A.8.3 VERSION-III: REVERSE-UNION

This version introduces a challenging cognitive workflow by intentionally obfuscating the query's entry points.

**Motivation and Design:** The Union method, while creating multi-source tasks, has a vulnerability: an agent could solve it with simple keyword searches for each source, bypassing deeper reasoning. Reverse-Union inverts the information flow, forcing an agent to first deduce a core 'anchor' entity (a second-layer entity) from descriptive clues and then use that entity as a pivot to expand its search.

**Example Question:** *Who are the authors from the same country as the 1980s prize-winner that wrote a novel about a group of British boys stranded on an uninhabited island, and who have also won both this reward and the Booker Prize? For each of them, what is their name, country, and the respective years they won each award?*

**Construction Process:** The construction builds upon the unified space from Version-II with a 'reverse' logic:

- **Source:** We use the unified information space from the Nobel and Booker prize union.

- **Select Anchor:** An entity at the intersection of the second layers is chosen as the 'anchor,' e.g., *William Golding*.

- **Obfuscate Anchor:** Instead of naming the anchor, unique descriptive clues based on its third-layer attributes are generated: "the 1980s prize-winner" and "wrote a novel about... British boys..." These clues become the 'Question Entities'.

- **Create Union Trigger:** A third-layer attribute of the anchor, his nationality (*British*), is selected as the pivot for the next stage of the query.

**Required Reasoning Process:** To solve this task, an agent must execute a two-stage process:

- **Deduction Stage:** The agent must first resolve the descriptive clues (which are third-layer entities) to identify the second-layer anchor entity. The clues "1980s prize-winner" and "novel about stranded British boys" uniquely point to *William Golding*. This inferential step is crucial.

- **Union Stage:** Having deduced William Golding, the agent identifies his nationality (a third-layer entity in his subtree): *British*. This becomes the pivot for the main query. The agent must then find all second-layer entities who (1) share this third-layer attribute (*British*) and (2) have won both the Nobel Prize and the Booker Prize. This requires filtering the unified entity space to find the final set of 'Target Entities', which includes authors like *William Golding*, *Kazuo Ishiguro*, and *J.M. Coetzee*.

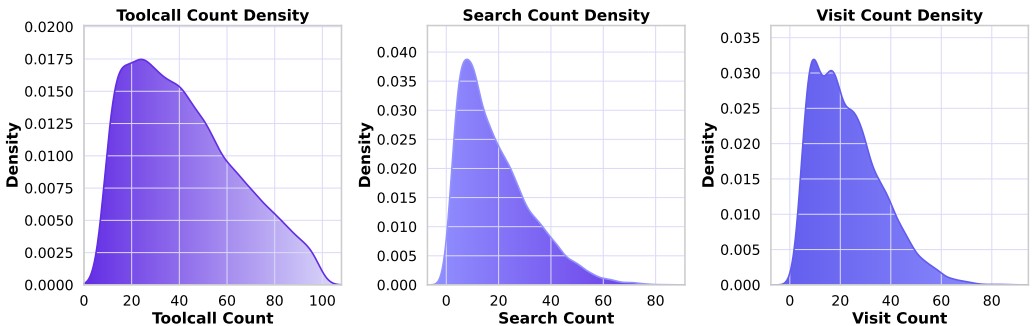

Figure 7: Distribution of *Search*, *Visit*, and total tool call.

### A.9 TOOL CALL ANALYSIS

As shown in Figure 7, our method involves a significantly large number of actions, including *Search*, *Visit*, and total tool calls. The density distributions indicate that tool calls often exceed several dozen per instance, with many cases surpassing 50 actions. This high frequency of actions reflects the intensive interaction and comprehensive exploration carried out by our approach, ensuring that the method thoroughly leverages available tools to achieve optimal performance.

### A.10 CLARIFICATIONS BETWEEN INFORMATION-SEEKING TASK AND ENTITY MINING

This section clarifies the scope of our entity-centric information-seeking (IS) framework and distinguishes it from mere "entity mining". We argue that modeling IS through an entity-centric lens is a reasonable, generalizable, and widely-adopted foundation for tackling complex IS tasks, rather than a restrictive simplification.

**Definition and Rationale of Entity-Centric IS** As formally defined in Section 2.1, an IS task is a tuple $T = \langle q, R \rangle$, where an agent interprets a natural language question $q$ and navigates the web to collect a complete set of required entities $R$. We define "entities" broadly to include not only final answers (e.g., "Alan Turing") but also intermediate reasoning stepping stones and attributes (e.g., "1950 Turing Award") that are critical for coherent inference.

This entity-centric task modeling is powerful for three core reasons: ENTITIES AS ATOMIC INFORMATION UNITS. Abstract questions (e.g., about events, procedures, or concepts) ultimately rely on grounding entities (e.g., "subprime mortgages" for the 2008 financial crisis) as building blocks for answers. UNAMBIGUOUS SEQUENTIAL REASONING. Discrete entities enable objective progress tracking, such as deducing anchor entities before retrieving related targets (a core mechanism in our Reverse-Union tasks). CROSS-TASK VERSATILITY. It generalizes to descriptive, causal, and procedural IS tasks, as all inherently involve collecting and connecting relevant entities.

**Community Consensus and Universality** Our entity-centric view aligns with a growing consensus in the field. Leading works such as WebResearcher (Qiao et al., 2025) model IS as collecting "reasoning entity chains" (analogous to our $R$), DeepDive (Lu et al., 2025) uses "entity coverage rate" (a precursor to our ISR), and industrial systems like MiroThinker (Team et al., 2025b) conceptualize IS as assembling "concept entity graphs". This paradigm is empirically validated and widely adopted for its effectiveness.

In all, the definition of information-seeking task is different from traditional entity mining. It's wildly adopted by current agent community.

### A.11 DISCUSSION OF PERFORMANCE ON MULTILINGUAL, OUT-OF-DISTRIBUTION, AND PRACTICAL SCENARIO

`WebLeaper`'s generalization capability in real-world scenarios is ensured through multiple inherent mechanisms and empirical validations: the agent architecture based on the ReAct framework (Yao et al., 2023) possesses inherent noise resilience, as its "thinking" stage prior to action enables the model to proactively assess the relevance and quality of retrieved information, identify and filter

redundant noise such as ads and irrelevant snippets, and dynamically adjust search strategies to adapt to the messy nature of the real web.

Centered on real-world and multilingual applicability, our experimental design adopts five benchmarks constructed from real web content—BrowserComp (Wei et al., 2025), GAIA (Mialon et al., 2023), Seal-0 (Pham et al., 2025), WideSearch (Wong et al., 2025), and xbench-DeepSearch (Xbench-Team, 2025), which cover common real-world challenges including ambiguous expressions, truncated entities, and scattered information. Furthermore, a portion of tasks in xbench-DeepSearch and WideSearch are presented in Chinese, making these benchmarks multilingual. Most of the selected benchmarks are not derived from Wikipedia, resulting in an out-of-distribution (OOD) evaluation setting. This helps mitigate bias in our synthetic data, as the evaluation is decoupled from the source of our training data synthesis. WebLeaper consistently outperforms strong baselines across these benchmarks, providing direct and robust empirical evidence for its generalization ability.

Furthermore, the dual-metric trajectory filtering mechanism of ISR (Information-Seeking Rate) and ISE (Information-Seeking Efficiency) further enhances the model's robustness: ISR requires the model to fully cover target entities, fostering its persistence and thoroughness in mining key information amid noise, while ISE suppresses ineffective "noise-chasing" behaviors and trains the model to prioritize high-value actions. Collectively, these designs ensure that WebLeaper can effectively excel at complex, noisy real-world application scenarios.

