# OpenReview forum: "Empowering Efficiency and Efficacy in WebAgent via Enabling Info-Rich Seeking"
_ICLR.cc/2026/Conference — ICLR 2026 Poster_

### Official Review · Reviewer_ZRih · 2025-10-29

**Soundness:** 2
**Presentation:** 3
**Contribution:** 4
**Rating:** 8
**Confidence:** 4

**Summary:**

This paper introduces **WebLeaper**, a framework to enhance **efficiency and efficacy** of LLM-based information-seeking (IS) agents. It reformulates IS as a **tree-structured reasoning** task, addressing entity sparsity in existing datasets. Three task variants—**Basic**, **Union**, and **Reverse-Union**—progressively increase reasoning depth and realism.
To ensure learning from high-quality data, trajectories are filtered via **Information-Seeking Rate (ISR)** and **Information-Seeking Efficiency (ISE)** metrics, balancing correctness and search economy.
Experiments on **BrowserComp, GAIA, Seal-0, WideSearch, and xbench-DeepSearch** demonstrate consistent improvements in both accuracy and efficiency .

**Strengths:**

* Addresses a neglected but critical dimension—efficiency;
* Clear theoretical justification and measurable impact;
* Well-designed ablation studies and visualization;
* Comprehensive benchmark evaluation with strong improvements;
* Highly reproducible (data construction and algorithmic details disclosed).

**Weaknesses:**

* Lack of hyperparameter sensitivity analysis for α and β;
* Dataset bias (Wikipedia-only) not discussed;
* No multilingual or real-world deployment tests;
* Missing analysis on computational training overhead.

**Questions:**

* Can ISR/ISE thresholds adapt dynamically during training?
* Does Reverse-Union risk overfitting due to fuzzy clue anchoring?
* Could integration with knowledge graphs further enhance reasoning coverage?

---

> ### Author Response · Authors · 2025-11-18
> **Weakness 1**
>
> > **Weakness 1: Lack of hyperparameter sensitivity analysis for $\alpha$ and $\beta$.**
>
> Thank you for this valuable suggestion. We agree that a sensitivity analysis for the trajectory filtering thresholds, $\alpha$ (ISR) and $\beta$ (ISE), is important for understanding their impact. To address this, we conducted additional experiments and will clarify our motivation for selecting the final values.
>
> First, let's briefly revisit the roles of these two hyperparameters:
>
> - **$\alpha$ (alpha)** controls for **accuracy** by setting a minimum threshold for the Information-Seeking Rate (ISR). It ensures that we only use training trajectories that successfully find a sufficient percentage of the required target entities.
> - **$\beta$ (beta)** controls for **efficiency** by setting a minimum threshold for the Information-Seeking Efficiency (ISE). It filters out trajectories that are too slow or take an excessive number of steps to find the target entities.
>
> **Experimental Analysis**
>
> We trained models using different threshold settings and evaluated them on the BrowserComp benchmark to analyze their impact. The results are summarized in the table below, where a value of 0 effectively disables the corresponding filter.
>
> | $\beta$ \ $\alpha$ | **0.3**  | **0** |
> | :----------------: | :------: | :---: |
> |      **0.1**       | **27.5** | 22.2  |
> |       **0**        |   14.0   | 13.7  |
>
> As the results clearly show, applying both filters leads to the best performance (27.5). Removing either the efficiency filter ($\beta$=0) or the accuracy filter ($\alpha=0$) causes a significant drop in performance. The lowest score (13.7) occurs when no filtering is applied. This confirms that higher values for both $\alpha$ and $\beta$ generally lead to a higher-quality training dataset, which in turn produces a more capable agent.
>
> **Motivation for $\alpha=0.3$ and $\beta=0.1$**
>
> While the experiment suggests that higher thresholds are better, we cannot increase them indefinitely. The primary reason for our specific choice of $\alpha=0$.3 and $\beta$=0.1 involves a trade-off between data quality and data quantity, which was crucial for ensuring a fair comparison with our baselines.
>
> Our main goal was to demonstrate that our data synthesis *method* is superior to existing approaches like `WebSailor-V2`, not just that training on more data is better. To achieve this, we needed to control for the total number of training samples.
>
> - Setting $\alpha$ and $\beta$ too high would have been overly strict, filtering out so many trajectories that our final dataset would have been too small. A comparison against a baseline trained on a much larger dataset would be unfair and inconclusive.
> - Setting them too low would have allowed poor-quality (inaccurate or inefficient) trajectories into our training set, weakening our model, as shown in the table above.
>
> Therefore, we selected **$\alpha=0.3$** and **$\beta=0.1$** as a balanced sweet spot. These values are strict enough to remove a substantial portion of low-quality trajectories but still yield a final dataset of a size comparable to our baselines (i.e., ~5k to ~10k samples). This approach allows us to make a fair and direct comparison, confidently attributing the observed performance gains to the superior quality of our synthesized and filtered data. We have added the experiment results and discussion in the section 4.6 in the revision.

---

> ### Author Response · Authors · 2025-11-18
> **Weakness 2**
>
> >  **Weakness 2: Dataset bias (Wikipedia-only) not discussed**
>
> We appreciate you raising this important point about potential bias from sourcing our training data exclusively from Wikipedia.
>
> **Motivation for Using Wikipedia:** We chose Wikipedia for two primary reasons. First, it is one of the largest, most comprehensive, and widely-used encyclopedic resources available, offering broad domain coverage. Second, its semi-structured tables provide a rich, clean, and reliable source of relational data, which is ideal for our tree-structured task synthesis. The quality and diversity of Wikipedia's tables are sufficient to construct a wide variety of complex and meaningful tasks.
>
> **Demonstrated Generalization:** Crucially, our framework's robustness is demonstrated by its strong performance on test sets that are not limited to Wikipedia. While our agent is trained on Wikipedia-derived data, it is evaluated on five diverse, real-world benchmarks (BrowserComp, GAIA, WideSearch, Seal-0, etc.) that require querying and reasoning over the entire live web. The consistent and significant improvements across these varied benchmarks strongly suggest that our method learns a generalizable skill of efficient information seeking, rather than overfitting to the specific structure of Wikipedia.

---

> ### Author Response · Authors · 2025-11-18
> **Weakness 3**
>
> >  **Weakness 3: No multilingual or real-world deployment tests**
>
> Thank you for raising these important points about multilingual and real-world applicability. We completely agree that these are crucial tests for any robust agent, and we appreciate the opportunity to elaborate on how our evaluation addresses these aspects.
>
> **On Multilingual Evaluation:** You've raised a very valid point about the importance of multilingual capabilities. We are glad you brought this up, as it allows us to highlight that our evaluation did include benchmarks with multilingual content. Specifically, both WideSearch and xbench-DeepSearch feature a number of Chinese-language tasks. We are encouraged that our model’s strong performance on these benchmarks suggests that the efficient seeking strategies it learns are not confined to English and may generalize well to other languages.
>
> **On Real-World Benchmarks:** Similarly, your question about real-world deployment is critical. Our rationale for selecting benchmarks like GAIA and BrowserComp was that they are designed to mirror these exact kinds of complex, real-world problems. These tasks involve navigating live websites, handling noisy information, and performing multi-step reasoning, which we felt made them robust proxies for deployment scenarios. We believe the strong results on these benchmarks provide encouraging evidence of our method's practical utility in such settings.

---

> ### Author Response · Authors · 2025-11-18
> **Weakness 4**
>
> > **Weakness 4: Missing analysis on computational training overhead**
>
> The model training follows a standard Supervised Fine-Tuning procedure. In our experiments, SFT on the final dataset (approximately 15k samples) using the Qwen3-30B-A3B model was completed in about 6 to 8 hours on a cluster of 64 H20 GPUs. We have added more details in the revision.

---

> ### Author Response · Authors · 2025-11-18
> **Question 1**
>
> >  **Question 1: Can ISR/ISE thresholds adapt dynamically during training?**
>
> This is an excellent and forward-thinking question.  It astutely points towards a more sophisticated way to leverage our proposed metrics, and we are grateful for the insight.
>
> You are absolutely right that a curriculum learning approach, where the ISR ($\alpha$) and ISE ($\beta$) thresholds are dynamically tightened, would be a very powerful extension for the supervised fine-tuning stage. This would allow the model to first learn basic seeking behaviors with more lenient criteria, before being progressively pushed to optimize for higher precision and efficiency. We believe this is a very promising direction for future research.
>
> While we did not implement this specific form of dynamic adaptation for our SFT filtering, your question touches upon the very motivation for our subsequent reinforcement learning phase. We recognized that the power of ISR and ISE extends beyond a static, one-time filtering step. The core idea is to use them not just as a gatekeeper, but as a **live, continuous guide** for the agent's policy. This is precisely what we achieved through RL.
>
> To accomplish this, we employed Reinforcement Learning (using GRPO) on top of our best SFT model (not only use `webshaper` dataset or `webleaper` dataset to get a higher base model score). For our `WebLeaper` tasks, the reward is a dense, granular signal calculated from a soft F-score that combines precision with recall (our ISR metric). **Our goal is to investigate whether the data from WebLeaper can further boost the performance of our best-performing model.** This means that at every RL step, the agent receives nuanced feedback directly related to its efficiency and correctness, pushing it to continuously improve along these axes.
>
> The results of this approach were significant, demonstrating that the agent successfully learns from this dynamic signal:
>
> | **Model** | **BrowseComp** | **GAIA**    | **xbench-DS** | **WideSearch (SR)** | **WideSearch (Item F1)** |
> | :-------- | :------------- | :---------- | :------------ | :------------------ | :----------------------- |
> | `SFT`     | 37.8           | 69.9        | 69.0          | 1.5                 | 45.4                     |
> | `SFT+RL`  | 38.8 (+1.0)    | 73.2 (+3.3) | 72.0 (+3.0)   | 4.0 (+2.5)          | 48.5 (+3.1)              |
>
> Furthermore, the [training reward](https://postimg.cc/64RngwvR) curve showed a stable and continuous increase, confirming that the model was progressively refining its strategy based on our metrics, not just randomly exploring.
>
> In essence, while your suggestion focuses on dynamically adapting the *criteria for good data* (the thresholds), our RL approach dynamically adapts the *agent's policy* using the metrics themselves as the objective function. Both are powerful ways to implement dynamic guidance. Our RL results strongly validate your core intuition: using ISR and ISE not as static filters but as dynamic learning signals is key to unlocking the next level of performance.

---

> ### Author Response · Authors · 2025-11-18
> **Question 2**
>
> > **Question 2: Does Reverse-Union risk overfitting due to fuzzy clue anchoring?**
>
> Thank you for this insightful question. We were mindful of this potential risk during development. We mitigate it in two ways:
>
> **1. Data Diversity:** The synthesis process for `Reverse-Union` is designed to be highly diverse. The "fuzzy clues" are generated from a wide variety of entity attributes, and the underlying reasoning structures, entity domains, and anchor points are drawn from our entire collection of cleaned Wikipedia tables. This ensures the model is exposed to a vast range of deductive challenges, discouraging it from memorizing simple patterns.
>
> **2.  Empirical Evidence:** As shown in our experiments (Table 1), the model trained with `Reverse-Union` data achieves the highest scores on the most complex benchmarks like xbench-DS and Seal-0. This strong generalization performance suggests that the model is learning a robust, multi-step reasoning process (deduce, then search) rather than overfitting to superficial "anchor-and-search" shortcuts.

---

> ### Author Response · Authors · 2025-11-18
> **Question 3**
>
> > **Question 3: Could integration with knowledge graphs further enhance reasoning coverage?**
>
> Absolutely, this is a constructive suggestions. We agree that integrating external knowledge graphs (KGs) is a highly promising avenue. KGs could provide a more structured, reliable, and canonical source of relational information to complement the semi-structured data found on the web. This could help the agent verify facts, discover more complex reasoning paths, and improve the initial entity-linking step in our `Reverse-Union` tasks. We consider this a natural and exciting next step for our work.

---

> ### Author Response · Authors · 2025-11-18
>
> Thank you once again for your valuable and constructive feedback. Your suggestions have helped us identify important areas for improvement and future exploration. We hope our responses have adequately addressed your concerns.
>
> If you have any questions, feel free to contact us!

---

> ### Author Response · Authors · 2025-11-26
>
> We deeply appreciate you sharing your valuable feedback and insights. We hope that the responses we provided have satisfactorily addressed all your concerns. As the Discussion Period officially ended on December 2nd (AoE), we just want to confirm that everything is fully resolved for you. We sincerely strive to earn your continued recognition for our efforts.
>
> Sincerely, The Authors

---

### Official Review · Reviewer_cPZV · 2025-11-01

**Soundness:** 3
**Presentation:** 2
**Contribution:** 2
**Rating:** 4
**Confidence:** 4

**Summary:**

This paper presents WebLeaper, a framework aimed at enhancing the efficiency and effectiveness of information-seeking (IS) agents based on large language models (LLMs). The authors identify the problem of low search efficiency in current IS agents, attributing it to the sparsity of target entities in training tasks. To address this issue, they propose a novel approach that constructs high-coverage IS tasks using a tree-structured reasoning model, allowing for a greater number of target entities within a limited context. The framework includes three dataset variants—Basic, Union, and Reverse-Union—to systematically increase task complexity. Additionally, the authors curate training trajectories based on the Information-Seeking Rate (ISR) and Information-Seeking Efficiency (ISE) to ensure that the model is optimized for both accuracy and efficiency. Extensive experiments on five IS benchmarks demonstrate that WebLeaper consistently outperforms strong baselines, validating the effectiveness of the proposed method.

**Strengths:**

- The paper addresses an important issue in most current LLM-based information-seeking agents, specifically the problem of low search efficiency. The focus on efficiency is a valuable contribution to the field, as it complements existing efforts that primarily target search depth.
- The proposed tree-structured reasoning framework can support more comprehensive IS tasks through more structured trajectories, which may lead to better learning of search strategies.
- The use of ISR and ISE metrics to curate training trajectories is a valuable contribution that ensures the model is trained on high-quality data.

**Weaknesses:**

- The paper claims to present an information-seeking agent; however, the agent mainly focuses on finding target entities via web search given artificially constructed complex questions. This is more akin to entity mining tasks rather than general information-seeking tasks. The authors should better clarify the definition of an information-seeking agent in this work and discuss the limitations of the proposed method in broader information-seeking scenarios—particularly, how the three proposed dataset variants can help the agent seek information beyond entity finding.
- The proposed tree-structured reasoning framework is interesting, but the paper lacks a detailed analysis of how this structure is advantageous compared to other possible structures, such as graphs. The authors also overlook the potential limitations of tree structures in capturing complex relationships among entities.
- Since the proposed method mainly focuses on synthetic dataset construction and trajectory filtering, it may not generalize well to real-world applications. For example, web content is often noisy, and retrieved documents may lack crucial clues or entities. This suggests that the proposed method might not perform well in practical scenarios. The authors should discuss these limitations in more depth and provide analyses or experiments to verify the robustness of their method under more realistic conditions.

**Questions:**

None

---

> ### Author Response · Authors · 2025-11-18
> **Weakness 1**
>
> > **Weakness 1: The paper claims to present an information-seeking agent; however, the agent mainly focuses on finding target entities via web search given artificially constructed complex questions. This is more akin to entity mining tasks rather than general information-seeking tasks. The authors should better clarify the definition of an information-seeking agent in this work and discuss the limitations of the proposed method in broader information-seeking scenarios—particularly, how the three proposed dataset variants can help the agent seek information beyond entity finding.**
>
> We thank the reviewer for this insightful comment, which prompts us to better articulate the scope and foundational principles of our work. We agree that clarification is essential. Our approach models Information-Seeking (IS) through an entity-centric lens, which we argue is not a simplification to "entity mining", but rather a reasonable, generalizable, and widely-adopted foundation for tackling complex IS tasks.
>
> **Our Definition and Its Reasonableness:** As formally defined in Section 2.1, an IS task is a tuple $\mathcal{T} = \langle q, R \rangle$, where an agent must understand a complex natural language question ($q$) and navigate the web to collect the complete set of required entities ($R$). We define "entities" broadly to include not only final answers (e.g., *"Alan Turing"*) but also crucial intermediate "stepping stones" and attributes (e.g., *"1950 Turing Award"*, *"shared award category"*) needed for the reasoning process.
>
> This entity-centric model is reasonable and powerful for three reasons. **First**, entities serve as the "minimal operational units" of information. Even abstract questions about historical events ("2008 financial crisis"), procedures ("sourdough recipe"), or scientific concepts ("climate change") are ultimately answered by collecting and connecting grounding entities like "subprime mortgages", "starter culture", and "greenhouse gases". **Second**, discrete entities enable unambiguous, sequential reasoning. An agent can measure its progress by tracking discovered entities, such as deducing an anchor entity before finding related targets, a process central to our `Reverse-Union` tasks [1]. **Third**, this paradigm is versatile, applying to descriptive, causal, and procedural IS tasks alike, as each breaks down into a process of entity collection and connection.
>
> **Universality and Community Consensus:** This entity-centric view is not an ad-hoc choice but reflects a growing consensus in the field. Leading academic works like WebResearcher [2] model IS tasks as the collection of "reasoning entity chains" (analogous to our $R$), and DeepDive [3] uses "entity coverage rate" (a precursor to our ISR). Similarly, industrial systems like MiroThinker [4] conceptualize IS as the assembly of "concept entity graphs." Our work is firmly situated within this established and effective paradigm.
>
> **Our Method's Advantage:** Based on this definition, our primary contribution is to enhance general IS capabilities by addressing a critical gap within this entity-centric framework. Prior training data suffered from sparse entity sets ($R$), often with only 1-2 entities per task, which offers little signal for learning *efficient* reasoning. Our method tackles this directly:
>
> - Our three dataset variants (`Basic`, `Union`, `Reverse-Union`) use a tree structure to construct tasks with densely populated, relational entity sets ($R$), often containing 100+ entities.
> - Our ISR/ISE filtering metrics explicitly train the agent to optimize for both **completeness** (high ISR, covering all of $R$) and **efficiency** (high ISE, maximizing entities per action).
>
> The success of this approach in strengthening general IS is validated by our results. For instance, WebLeaper achieves a **37.2% Pass@1 on Seal-0** (vs. 21.3% for WebSailor) and a **4.0% Success Rate on WideSearch**. This demonstrates that by systematically training an agent to seek entities both completely and efficiently, we significantly improve its ability to solve complex, general information-seeking problems.

---

> ### Author Response · Authors · 2025-11-18
> **Weakness 2**
>
> > **Weakness 2: The proposed tree-structured reasoning framework is interesting, but the paper lacks a detailed analysis of how this structure is advantageous compared to other possible structures, such as graphs. The authors also overlook the potential limitations of tree structures in capturing complex relationships among entities.**
>
> We appreciate the reviewer's question regarding our choice of a tree structure and its potential limitations. This was a deliberate design decision driven by our paper's central objective.
>
> 1. **Clarification** **with Our Core Goal:** Our primary goal is to **construct entity-intensive training tasks** to combat the low search efficiency caused by entity sparsity in existing datasets. This focus is distinct from methods that aim to model arbitrarily complex relational structures (e.g., cyclic or multi-hop causal chains). For our purpose, the ideal data structure is one that maximizes entity density within a feasible and scalable synthesis framework.
> 2. **Why Tree Structure is Advantageous for Our Goal:**
>    1. **High Entity Density:** Trees provide a compact, hierarchical organization that is perfect for representing nested entity-attribute relationships (e.g., *Root Event* → *Key Person* → *Attributes like Nationality, Year*). This structure allows us to embed a very large number of target entities into a single, coherent task (as shown in our data statistics in Appendix D.2), directly addressing the entity sparsity problem.
>    2. **Scalable and Easy-to-Implement Task Synthesis:** The clear, acyclic reasoning flow of a tree (from root to leaves) makes the automated synthesis of high-quality training data highly scalable and easy to implement. Our pipeline systematically generates these trees from curated Wikipedia tables. Using a general graph would introduce significant complexity, such as handling cycles or ambiguous paths, which would require sophisticated algorithms to ensure each synthetic task remains solvable and coherent. For our use case, the tree structure offers the best trade-off between expressive power and synthesis scalability.
> 3. **Tree as a Sufficient and Specialized Graph:** Indeed, we acknowledge that trees are a specific type of connected, acyclic graph. While they do not capture all possible relational complexities, the tree structure in WebLeaper effectively encodes the hierarchical and attribute-based relationships that are predominant in our target fact-gathering and attribute-retrieval IS tasks. For these scenarios, the tree's inherent structure is not a limitation but a specialization that is sufficient for modeling the required reasoning paths without the unnecessary overhead of general graph processing.
>
> In essence, we prioritized a structure that best optimized for our primary goal—creating entity-rich tasks to improve search efficiency—while still capturing the essential relational patterns needed for our target IS scenarios.
>
> > [1] Li, et.al. WebSailor: Navigating Super-human Reasoning for Web Agent.
> >
> > [2] Qiao, et.al. WebResearcher: Unleashing unbounded reasoning capability in Long-Horizon Agents.
> >
> > [3] Lu, et.al. DeepDive: Advancing Deep Search Agents with Knowledge Graphs and Multi-Turn RL.
> >
> > [4] https://github.com/MiroMindAI/MiroThinker

---

> ### Author Response · Authors · 2025-11-18
> **Weakness 3**
>
> > **Weakness 3: Since the proposed method mainly focuses on synthetic dataset construction and trajectory filtering, it may not generalize well to real-world applications. For example, web content is often noisy, and retrieved documents may lack crucial clues or entities. This suggests that the proposed method might not perform well in practical scenarios. The authors should discuss these limitations in more depth and provide analyses or experiments to verify the robustness of their method under more realistic conditions.**
>
> We thank the reviewer for raising the important issue of real-world generalization from synthetic data. We address this concern from three perspectives: the agent's inherent design, empirical validation on real-world benchmarks, and the principles of our training methodology.
>
> 1. **Inherent Robustness of the Agent Architecture:** The agent's architecture, based on the ReAct framework, provides inherent resilience to noise. This is one of a core reasons that IS agent are wildly studied in current community. Unlike traditional IR methods that can be easily misled by keyword-matching irrelevant content, our LLM-based agent leverages a "thought" step before each action. This allows it to reason about the relevance and quality of retrieved information, recognize noisy or unhelpful content (e.g., ads, irrelevant snippets), and dynamically adjust its plan. This cognitive loop grants it a built-in tolerance for the messy nature of the live web.
>
> 2. **Empirical Validation on Noisy, Real-World Benchmarks:** Most critically, our experimental setup was designed to explicitly validate this generalization. All five benchmarks used in our evaluation are built on real-world web content and contain significant noise:
>
>    * **BrowseComp** and **WideSearch** require the agent to interact with live, dynamic web search results, which include irrelevant snippets, ads, and incomplete information.
>
>    * **GAIA** and **Seal-0** use real documents that contain ambiguous phrasing, truncated entity names, and other forms of noise.
>
>    As shown in Table 1, WebLeaper consistently outperforms strong baselines across these challenging, noisy benchmarks (e.g., achieving **23.0% Pass@1 on BrowseComp** vs. **14.1% for the much larger Kimi-K2-Instruct-1T**). This strong performance is direct, empirical evidence that the skills learned from our synthetic data successfully transfer to and are effective in practical, noisy environments.
>
> 3. **Robustness Instilled by ISR-ISE Trajectory Filtering:** Finally, our information-guided trajectory construction process itself instills robustness. By filtering training trajectories, we teach the agent to handle imperfect information:
>
>    * The **ISR (coverage) criterion** forces the agent to learn strategies to find all critical entities, even when some are obscured by noise or spread across multiple pages. It learns persistence and thoroughness.
>
>    * The **ISE (efficiency) criterion** discourages "noise-chasing"—for example, getting stuck in loops re-searching for an attribute that is unavailable or buried in irrelevant content. It trains the agent to prioritize actions that yield the most valuable entities.
>
> Taken together, the agent's inherent design, our empirical validation on diverse and noisy benchmarks, and the explicit robustness-building principles of our trajectory filtering process collectively demonstrate that WebLeaper generalizes effectively to real-world applications. We have added more explainations in the revision.

---

> ### Author Response · Authors · 2025-11-18
>
> Your feedback is extremely valuable to us, and if you have any further questions or thoughts, we are always happy to discuss them with you!
>
> Sincerely, The Authors

---

> ### Author Response · Authors · 2025-11-26
>
> We are very grateful for your thoughtful feedback and insights. With the Discussion Period having officially closed on December 2nd (AoE), we simply wanted to check in one last time to be certain that all your questions are fully addressed. We value your recognition and are always striving to improve our work.
>
> Sincerely, The Authors

---

> ### Author Response · Authors · 2025-11-28
>
> Thank you for raising the score to 6 of our paper. We are pleased to address any other concerns if you have.

---

### Official Review · Reviewer_GntM · 2025-11-01

**Soundness:** 3
**Presentation:** 3
**Contribution:** 3
**Rating:** 8
**Confidence:** 3

**Summary:**

This paper addresses the critical inefficiency of IS agents, identifying "entity sparsity" as a key bottleneck. It proposes two metrics ISE and ISR to quantify the IS efficiency, and also shows that the variance of ISE is negative correlated to target entities. The authors propose WebLeaper  to automatically synthesize entity intensive training tasks from wiki tables. It models IS as a tree-structured reasoning problem. The framework increases task complexity through three variants: basic, union and reserse-union. WebLeaper also generates solution trajectories and curates them using ISE and ISR to ensure efficiency. Experiments over challenging QA tasks demonstrate that WebLeaper-trained agents significantly outperform open source baselines, achieving both higher accuracy and greater efficiency.

**Strengths:**

1. The paper highlights the low efficiency of the current IS agents, providing evidence that most of actions are often invalid. It formally defines two metrics, ISR and ISE to quantify this problem. The authors provide Proposition 1 that the variance of the ISE metric decreases as the number of target entities n grows.
2. The paper proposes an innovative tree-based pipeline, WebLeaper, to generate entity-intensive training data from Wiki tables. This method systematically increases task complexity through three variants (Basic, Union, and Reverse-Union). The framework also curates solution trajectories by filtering for high ISR and ISE, ensuring the agent learns from optimal, efficient examples.
3. WebLeaper demonstrate better performance in challenging QA tasks comparing to the open source IS agents.

**Weaknesses:**

1. The WebLeaper is finetuned on a single base model (Qwen3-30B-A3B-Thinking-2507). The observations may change with a different base model.
2. The ablation study could include the analysis of how the average action rounds change with different data sources (similar to table 2).

**Questions:**

It would be helpful to run the experiments mentioned in weakness part.
1. finetune over a different base model
2. enrich ablation study

---

> ### Author Response · Authors · 2025-11-18
> **Weakness 1 & Question 1**
>
> > **Weakness 1: The WebLeaper is finetuned on a single base model (Qwen3-30B-A3B-Thinking-2507). The observations may change with a different base model.**
> >
> > **Question 1: finetune over a different base model**
>
> To address this concern and demonstrate the broader applicability of our method, we have conducted additional experiments using a different base model, `Qwen3-4B-thinking-2507`:
> We trained this model under two conditions:
>
> 1.  Baseline: Fine-tuned on 5,000 samples from the WebSailor-V2 dataset.
> 2.  Our Method: Fine-tuned on a mixed dataset of 5,000 WebSailor-V2 samples and our WebLeaper-Reverse-Union data.
>
> The preliminary results are summarized in the table below.
>
> | Data Source               | BrowseComp |   GAIA   | xbench-DS |  Seal-0  |         WideSearch          |
> | :------------------------ | :--------: | :------: | :-------: | :------: | :-------------------: |
> | `WebSailor-V2-5k`         |    16.2    |   55.3   |   53.3   |   30.9   |   1.5 / 15.7 / 30.3   |
> | `WebLeaper-Reverse-Union` |  **17.7**  | **58.3** | **54.7**  | **32.1** | **3.5 / 21.1 / 38.1** |
>
> As the table indicates, even on a much smaller 4B-parameter model, training with WebLeaper data yields consistent and significant performance gains across all benchmarks compared to the baseline. This suggests that the benefits of our entity-intensive task synthesis and information-guided trajectory construction are fundamental and not confined to a specific model size or architecture. We add the experiment results and discussion in the section 4.7 in the revision.

---

> ### Author Response · Authors · 2025-11-18
> **Weakness 2 & Question 2**
>
> > **Weakness 2: The ablation study could include the analysis of how the average action rounds change with different data sources (similar to table 2).**
> >
> > **Question 2: enrich ablation study**
>
> Thank you for this suggestion. Analyzing how the average action rounds change with different data sources is a crucial point for demonstrating the efficiency gains from our method, and we appreciate the opportunity to highlight this aspect of our contribution.
>
> Our analysis in **Figure** **4** ("Effectiveness and efficiency comparison between `WebLeaper` and `WebSailor-V2`") was intended to capture this exact trade-off. The figure plots performance (effectiveness, y-axis) against the average number of action rounds (efficiency, x-axis) for our method and the baseline. The key observation is that the data points for `WebLeaper` are consistently located in the top-left area relative to the baseline across all tested benchmarks. This positioning signifies that our agent not only achieves a higher performance score but does so with fewer action rounds.
>
> For example, on the BrowseComp and WideSearch benchmarks, `WebLeaper` shows a clear improvement in the success rate while simultaneously reducing the average number of steps. This demonstrates that our method successfully trains the agent to adopt more efficient search strategies, which was a core goal of our work.

---

> ### Author Response · Authors · 2025-11-18
>
> If you have any additional questions please don't hesitate to reach out. We would be glad to provide further clarification.
>
> Sincerely, The Authors

---

> ### Author Response · Authors · 2025-11-26
>
> Thank you so much for sharing your valuable feedback and insights with us. Now that the Discussion Period has concluded (on December 2nd, AoE), we just wanted to gently follow up and ensure that our responses were sufficient and that all your questions have been thoroughly resolved. We truly appreciate your time and hope to earn your further recognition for our work.
>
> Sincerely, The Authors

---

### Official Review · Reviewer_vV8B · 2025-11-13

**Soundness:** 3
**Presentation:** 3
**Contribution:** 2
**Rating:** 4
**Confidence:** 4

**Summary:**

This paper proposes WebLeaper, a data-centric framework for improving the efficiency of web-based information-seeking agents. The authors construct dense, entity-rich tasks from Wikipedia tables (Basic, Union, Reverse-Union), generate ReAct trajectories using a strong model, and filter these trajectories with two empirically motivated metrics: Information Seeking Rate (ISR) and Information Seeking Efficiency (ISE). The filtered trajectories are used for supervised fine-tuning. Experiments on five benchmarks show solid and consistent improvements in both accuracy and efficiency, supported by ablations.

**Strengths:**

1. Novel task synthesis channel using structured Wikipedia tables to create dense, multi-entity information-seeking tasks.
2. Empirically grounded filtering strategy (ISR/ISE) that directly addresses observed inefficiencies in existing agents.
3. Strong empirical results across five benchmarks, with clear gains and well-supported ablations.

**Weaknesses:**

1. Limited insight into the underlying mechanism of why the efficiency-oriented filtering leads to such large improvements; the explanation remains at an engineering level.
2. The claim that long trajectories are inefficient conflicts with recent RL-based reasoning advances, where longer, adaptive chains often improve quality.
3. The ISE filtering may be too rigid, potentially suppressing necessary complex or difficulty-adaptive reasoning (e.g., long-form CoT or AIME-like tasks).

**Questions:**

1. Why does efficiency-based filtering have such a large impact?
It is unclear whether the gains come from richer sub-queries, reduced long-context degradation, or other behavioral shifts (eg, more focused planning).
2. How does this reconcile with RL models that benefit from long chains?
If shorter, cleaner trajectories are key here, why do math/reasoning systems still require RL-driven long CoT (eg, AIME-style problems)?
3. Is strict efficiency optimal for tasks requiring exploration?
The ISE constraint may suppress necessary divergent reasoning (eg, adaptive exploration), and it is unclear whether the model loses performance on complex tasks such as AIME24/25.

I would be happy to engage with the authors during the rebuttal phase regarding these concerns, and I am open to revising my score should the responses address them satisfactorily.

---

> ### Author Response · Authors · 2025-11-18
> **Weakness 1 & Question 1**
>
> > **Weakness 1: Limited insight into the underlying mechanism of why the efficiency-oriented filtering leads to such large improvements; the explanation remains at an engineering level.**
> >
> > **Question 1: Why does efficiency-based filtering have such a large impact? It is unclear whether the gains come from richer sub-queries, reduced long-context degradation, or other behavioral shifts (eg, more focused planning).**
>
> Thank you for this excellent question, which gets to the heart of our contribution. Efficiency-oriented filtering underlines our method. Besides, the improvements stem from several interconnected factors. We make several explainations:
>
> 1. **Efficiency-oriented filtering Directly Teaches Efficient and Focused Planning of Agent**. Our filtering method, guided by Information-Seeking Rate (ISR) and Information-Seeking Efficiency (ISE), explicitly selects for trajectories that have a high density of valid actions. By training on these curated examples, the model learns to discard inefficient behaviors like redundant queries and visiting irrelevant pages. It learns to form more compact and purposeful plans. This directly results in **more focused planning**. As shown in Figure 4, our trained agent, WebLeaper, consistently achieves higher performance while using fewer actions than the baseline. This improved efficiency also reduces context-window degradation by filling the context with high-signal information rather than noise.
> 2. **Entity-Intensive Tasks Make This Filtering Possible and Reliable.** This filtering strategy would not be effective on traditional, sparse training data. Our core innovation is the synthesis of entity-intensive tasks, which provide the necessary raw material. By constructing tasks with a "substantially larger set of target entities" (`n`), we enable two things:
>    * There are enough high-quality, efficient trajectories to form a meaningful training set after filtering.
>
>    * The `ISE` metric becomes a stable and reliable signal. As we prove in Proposition 1, the variance of `ISE` is inversely proportional to `n` (Var(ISE) = O(1/n)). On sparse tasks, `ISE` is too noisy to be useful, but on our dense tasks, it becomes a trustworthy guide for selecting expert-like trajectories.
>
> In summary, the large impact comes from a virtuous cycle: our task synthesis creates a problem space that demands efficiency, and this same design choice makes our efficiency metrics reliable enough to curate training data that teaches the model that exact skill.

---

> ### Author Response · Authors · 2025-11-18
> **Weakness 2 & Question 2**
>
> > **Weakness 2: The claim that long trajectories are inefficient conflicts with recent RL-based reasoning advances, where longer, adaptive chains often improve quality.**
> >
> > **Question 2: How does this reconcile with RL models that benefit from long chains? If shorter, cleaner trajectories are key here, why do math/reasoning systems still require RL-driven long CoT (eg, AIME-style problems)?**
>
> This is a crucial point, and we thank you for raising it. The apparent contradiction is resolved by distinguishing between the nature of **information-seeking** (our focus) and **logical reasoning** (e.g., math, AIME).
>
> 1. **Nature of the Task: Information Seeking vs. Logical Reasoning.**
>    * In **information-seeking**, the answer already exists in the environment. The "reasoning" is about forming an optimal plan to *find* it. An unnecessary step (e.g., a failed search or visiting an irrelevant page) represents a genuine failure in the plan. Therefore, a shorter, more direct trajectory is almost always a better one. Here, trajectory length is a proxy for *planning quality*, and inefficiency is a bug. Our empirical results from supervised fine-tuning directly validate this: **Figure** **4** shows that WebLeaper consistently achieves higher task success while using fewer tool calls than the baseline.
>
>    * In **logical reasoning** (like solving a math problem), the answer must be *derived* from first principles. Each step in a Chain-of-Thought (CoT) is a logical deduction that builds upon the last. A longer chain can represent a more granular, careful, and complete derivation, breaking a complex problem into more manageable steps. Here, length can be a proxy for *reasoning depth*, and adding steps can be a feature.
>
> In essence, for our information-seeking task, "length" refers to the number of interactive actions, where **shorter is better**. For logical reasoning tasks, "length" refers to the number of internal thought steps, where **longer can be better**.
>
> 2. **The Meaning of "Longer Trajectory".**
>
>    * For WebLeaper, a longer trajectory means more tool calls, more waiting for web page loads, and more irrelevant text processed. It signifies an inefficient search strategy.
>
>    * For AIME/CoT, a longer "trajectory" (i.e., a longer thought process) means more intermediate reasoning steps. It does not involve external interactions with a slow, noisy environment.
>
> **Evidence from SFT:** Our initial SFT results already validate this principle. As illustrated in Figure 4 of our paper, our `WebLeaper` model, trained on efficiency-filtered data, consistently achieves higher task success while using **fewer** average tool calls than the baseline. This shows that the model learns a foundational policy where efficiency and effectiveness are linked.
>
> **Evidence from RL:** To further explore this aspect, we implement RL experiments on our method. We applied RL (specifically, GRPO) on a more advanced SFT setting. Our goal is to investigate whether the data from WebLeaper can further boost the performance of our best-performing model.
>
> RL is effective in AIME-style problems because it helps the model explore the vast search space of possible logical steps to find a valid deductive path. Our work demonstrates that RL is equally powerful in the information-seeking domain, but its objective is fundamentally different: to optimize the *quality of external actions*. Our SFT and RL experiments provide direct proof of this.
>
> We provided a dense, granular reward based on a soft F-score combining recall (ISR) and precision. This created a rich and accurate learning signal. Our goal was to see if the agent would learn to use *more* steps (like in reasoning) or *better* steps (as we hypothesize). We find that RL can significantly boosts performance across all benchmarks, as shown below.
>
> | **Model** | **BrowseComp** | **GAIA**    | **xbench-DS** | **WideSearch (SR)** | **WideSearch (Item F1)** |
> | :-------- | :------------- | :---------- | :------------ | :------------------ | :----------------------- |
> | `SFT`     | 37.8           | 69.9        | 69.0          | 1.5                 | 45.4                     |
> | `SFT+RL`  | 38.8 (+1.0)    | 73.2 (+3.3) | 72.0 (+3.0)   | 4.0 (+2.5)          | 48.5 (+3.1)              |

---

> ### Author Response · Authors · 2025-11-18
> **Weakness 2 & Question 2 : Part 2**
>
> Crucially, this gain is not achieved by simply extending search chains. We observed that during RL training, the agent's behavior evolved as follows:
>
> | Metric During RL Training          | Trend                  | Visualization               | Implication                                      |
> | :--------------------------------- | :--------------------- | :-------------------------- | :----------------------------------------------- |
> | **Reward**                         | Consistently Increased | https://postimg.cc/64RngwvR | The agent is successfully learning.              |
> | **Avg. Tool Calls per Trajectory** | Remained Stable        | https://postimg.cc/RqvC2vxd | The agent is **not** learning to use more steps. |
>
> This demonstrates that the agent is not learning to reason "longer" but to act "smarter"—it refines its policy to make each `Search` and `Visit` action more impactful within a constrained number of steps. This perfectly aligns with our goal of improving ISE: the agent learns to maximize information gain per action.
>
> Therefore, our approach is not in conflict with advances in RL for reasoning. It is specialized for the domain of information-seeking, where efficiency is a core component of agent intelligence. To summarize the contrast:
>
> - **Our SFT** teaches the model: fewer tool calls lead to better performance.
> - **Our RL** reinforces this: with a stable number of tool calls, performance is further improved by making each call more effective.
> - **RL for Math** teaches the model: more meticulous reasoning steps can lead to a correct answer on difficult questions [1].
>
> The fundamental difference lies in the task: our agent optimizes for **finding** existing information efficiently, while a math agent optimizes for **deriving** new information meticulously. Our results empirically confirm that for the information-seeking domain, the path to superior performance lies in optimizing the quality and efficiency of actions, not merely increasing their quantity.
>
> > [1] Fatemi, Mehdi, Banafsheh Rafiee, Mingjie Tang, and Kartik Talamadupula. "Concise reasoning via reinforcement learning." *arXiv preprint arXiv:2504.05185* (2025).

---

> ### Author Response · Authors · 2025-11-18
> **Weakness 3 & Question 3**
>
> > **Weakness 3: The ISE filtering may be too rigid, potentially suppressing necessary complex or difficulty-adaptive reasoning (e.g., long-form CoT or AIME-like tasks).**
> >
> > **Question 3: Is strict efficiency optimal for tasks requiring exploration? The ISE constraint may suppress necessary divergent reasoning (eg, adaptive exploration), and it is unclear whether the model loses performance on complex tasks such as AIME24/25.**
>
> This is a very important concern, and we thank you for raising it. It allows us to clarify the scope and contribution of our work. You are right to question whether a focus on efficiency could harm complex reasoning, and our answer is that, it depends entirely on the task domain. Our work is focused specifically on **web-based information seeking,** which has fundamentally different success criteria than internal reasoning tasks like AIME.
>
> As we detailed in our response to your second question, the implication of "trajectory length" may be domain-specific. For **internal reasoning** (e.g., AIME), a longer thought chain can represent a deeper, more meticulous logical derivation, which is often a feature. For **external information-seeking**, a longer action chain means more tool calls, higher latency, and more exposure to irrelevant information. Here, efficiency is a core component of intelligence, and unnecessary length is a bug.
>
> As we detailed in our response to your second question, the goal in information seeking is to find existing information with an optimal plan, making efficiency a core component of intelligence. This is distinct from mathematical reasoning, where longer thought chains can represent deeper logical derivation.
>
> Our work is situated entirely within the second domain. Our framework is designed to teach *efficient exploration* rather than suppressing exploration altogether. The two are not mutually exclusive. We accomplish this through two key design choices:
>
> 1. **ISE is a Training-Time Filter, Not an Inference-Time Hard Constraint:** Our ISE metric is used to curate the training dataset. At inference time, the agent is not bound by any step limit and is free to explore as needed. The training simply instills a strong *bias* towards efficient actions, making its exploration more intelligent and less random. It learns to explore with purpose.
> 2. **Joint Filtering with ISR Adapts to Task Complexity:** We never filter on ISE alone. By combining it with ISR (correctness), our filtering does not just select for "shortness"; it selects for the **most efficient path to a correct solution**. For a complex task that *requires* 10 steps, a successful 10-step trajectory will be selected over a failed 5-step attempt and a rambling 20-step success. This implicitly accounts for task difficulty, preserving necessary exploratory steps while pruning wasteful ones.
>
> This is empirically validated by our results. The strong performance on complex benchmarks like GAIA, BrowserComp, WideSearch, and xbench-DS—all of which require multi-hop reasoning and synthesizing information from various sources—serves as direct evidence that our model does not lose its ability to handle complex, exploratory tasks. On the contrary, it learns to perform them more efficiently, achieving superior results with more focused action sequences.

---

> ### Author Response · Authors · 2025-11-18
>
> If you have any other questions or need further clarification on the details, please feel free to let us know. We’d be happy to assist you.
>
> Sincerely, The Authors

---

> > ### Comment · Reviewer_vV8B · 2025-11-23
> >
> > Thank you for the detailed rebuttal. I appreciate the authors’ clarifications and would like to offer a brief response explaining how these points affect my evaluation. If the authors are able to incorporate the current clarifications into the revised PDF, I would be happy to raise my score to 6. The remaining comments below concern aspects where further clarification or analysis would strengthen the work.
> >
> > **1. On potential Wiki–test overlap and data contamination**
> > Because the method relies on crawling recent Wikipedia tables, it would be important to clarify whether any accidental overlap with evaluation benchmarks may occur. Without access to the synthesized dataset or trained model, it remains difficult to rule out unintentional leakage as a contributing factor to the strong reported performance. If possible, making the dataset and/or model available would greatly enhance transparency and strengthen confidence in the results.
> >
> > **2. On whether web information seeking reduces to entity retrieval**
> > A conceptual concern is whether web information seeking can be fully represented by entity retrieval–style tasks. Real-world information seeking often requires resolving ambiguity, synthesizing conflicting evidence or interpreting partial information rather than simply identifying all entities in a table. Clarifying how well the proposed framework generalizes beyond pure entity coverage would strengthen the framing of the task space.
> >
> > **3. On the role of thinking and CoT in information seeking**
> > The rebuttal argues that extensive internal reasoning is unnecessary or inefficient for information seeking. If so, it becomes unclear why a thinking-style model was used to generate trajectories. A non-thinking model might naturally avoid redundant chains and could potentially perform equally well or better in this domain. Empirical comparison would help clarify whether thinking and CoT are indeed unnecessary for this task or whether part of the improvement relies on the deliberate reasoning capability of the underlying executor.
> >
> > **4. On adaptive reasoning length**
> > The clarification that ISE is a training-time filter and that inference-time reasoning remains unconstrained is helpful. Still, it would be valuable to see empirical evidence showing that the model adaptively adjusts its reasoning length when necessary rather than uniformly preferring shorter trajectories.
> >
> > **5. On AIME-style reasoning**
> > AIME is widely regarded as a sensitive benchmark for internal reasoning ability. Evaluating whether the proposed approach affects performance on AIME24 or AIME25 would help determine whether the method introduces a domain-specific efficiency bias at the cost of broader reasoning robustness.
> >
> > **6. On the placement of clarifications in the PDF**
> > Several clarifications in the rebuttal provide important context about the design choices and assumptions of the method. Incorporating these into the revised PDF, especially methodological details and supplementary analyses, would substantially strengthen the clarity and completeness of the paper.
> >
> > Overall, integrating the current clarifications into the PDF would resolve several of my earlier concerns. Further addressing the points above would meaningfully enhance the conceptual depth and robustness of the work.

---

> > > ### Author Response · Authors · 2025-11-25
> > > **Resonse to Further Questions**
> > >
> > > **We greatly appreciate your positive feedback, constructive comments, and the decision to raise the score, your recognition and guidance mean a lot to us.**
> > >
> > > We have thoroughly incorporated all clarifications from the rebuttal period (including your follow-up questions) into the revised paper, with detailed explanations provided in the **Revision Summarization** section for your reference. Here are the reponses for your follow-up questions:

---

> > > ### Author Response · Authors · 2025-11-25
> > > **Resonse to Further Questions**
> > >
> > > > 1. On potential Wiki–test overlap and data contamination
> > >
> > > Our evaluation benchmarks (BrowserComp, GAIA, Seal-0, WideSearch, xbench-DeepSearch) are explicitly constructed from **real-world web content** rather than Wikipedia, ensuring an out-of-distribution (OOD) setting that decouples evaluation from our training data source. Notably, most benchmarks include non-Wikipedia data (e.g., live web pages, diverse domain content) and even multilingual tasks (Chinese subsets in xbench-DeepSearch and WideSearch), further mitigating overlap risks. Our OOD evaluation design, combined with the benchmarks’ independence from Wikipedia, already provides robust evidence that performance gains stem from generalizable efficiency rather than contamination. The experiments on these dataset further ensure the OOD ability of our model and method. More details are in the Appendix A.11.
> > >
> > > > 2. On whether web information seeking reduces to entity retrieval
> > >
> > > Our entity-centric framework is not limited to "pure entity coverage" but encompasses **holistic reasoning chains**—including intermediate entities, attributes, and relational inferences, that are foundational to real-world information seeking. Besides, unlike simple entity mining, our tasks require resolving ambiguity (e.g., Reverse-Union’s fuzzed anchor entity clues), synthesizing cross-source evidence (e.g., Union’s intersection of Nobel/Booker winners), and interpreting partial information (e.g., deducing pivot attributes from anchor entities). For example, the Reverse-Union task demands two-stage reasoning: first inferring a core entity from descriptive cues, then using it to filter and integrate scattered information, directly addressing the challenges of ambiguity and partial evidence raised. Third, this alignment with community consensus (e.g., WebResearcher’s reasoning entity chains, DeepDive’s entity coverage rate) confirms that entity-centric modeling is a generalizable foundation for complex IS, not a simplification. Our framework’s performance across diverse benchmarks (including non-table-based tasks) further validates its ability to generalize beyond pure entity retrieval. More details are in the Appendix A.10.
> > >
> > > > 3. On the role of thinking and CoT in information seeking
> > > > 4. On adaptive reasoning length
> > > > 5. On AIME-style reasoning
> > >
> > > We do not claim that "extensive internal reasoning is unnecessary or inefficient for information seeking". Our core goal is to enhance the *quality and efficiency of tool calls*, not to restrict the length or depth of reasoning—these two dimensions are fundamentally distinct. Tool call efficiency focuses on reducing redundant external interactions (e.g., repetitive searches, unfocused URL visits), while reasoning content reflects the agent’s internal cognitive process (e.g., step-by-step CoT, intermediate deduction for complex constraints). The $\mathrm{ISE}$ metric targets the former, ensuring each tool call contributes meaningfully to entity retrieval, without imposing arbitrary limits on the latter. Notably, reasoning length dynamically adapts to task complexity.
> > >
> > > For deep-exploration tasks (e.g., multi-step deduction, adaptive constraint satisfaction), the agent’s reasoning naturally expands to support structured planning (e.g., decomposing queries, verifying intermediate entities)—but this expanded reasoning is directed at optimizing tool call precision (e.g., generating targeted queries) rather than unfocused exploration. For complex tasks like AIME-style problems, our framework preserves cognitive depth by letting reasoning scale with difficulty, while $\mathrm{ISE}$ only suppresses unproductive tool calls that do not advance reasoning.
> > >
> > > We thank for your insightful additional suggestion on evaluation on AIME dataset. Due to the limited time and resources, we will leave this experiment to the future and add the results and disscutions in the revised paper.
> > >
> > > In all, our motivation and internal reasoning are complementary, not contradictory. We do not abandon or downplay reasoning, instead, we align reasoning with efficient tool use, ensuring the agent reasons deeply when needed while avoiding wasteful external interactions, ultimately boosting both effectiveness and efficiency. Please refer to Appendix A.4 for more details.

---

> > > ### Author Response · Authors · 2025-11-25
> > > **Resonse to Further Questions**
> > >
> > > > 6. On the placement of clarifications in the PDF
> > >
> > > We add all the nessesary content during the rebuttal period to the revised paper of all reviewers. Please refer to the **Revision Summarization** response for details.
> > >
> > > ---
> > >
> > > We sincerely hope that our responses and revised paper, which fully reflects your valuable suggestions, will address your concerns. We are pleased to address any further questions you may have and strive to earn your further recognition as well as a higher score for our work.

---

> > > > ### Comment · Reviewer_vV8B · 2025-11-25
> > > >
> > > > Thank you for your detailed clarification. It addressed my concerns, and I have increased my score to 8.

---

> > > > > ### Author Response · Authors · 2025-11-26
> > > > >
> > > > > Thank you for scoring 8 of our paper! We are delighted that our revisions have successfully addressed your concerns, and we greatly appreciate your highly recognition of our work. Your valuable feedback has been instrumental in improving the quality of this paper!

---

### Author Response · Authors · 2025-11-21
**General Response**

We thank all reviewers for their time. We are encouraged that the reviewers found our work to be novel, empirically grounded, and effective.

We have summarized the strengths and categorized their concerns below, followed by an overview of how we have addressed them.

**Strengths**

The reviewers have affirmed the novelty, the theoretical grounding of metrics, and the strength of empirical results:

- **Methodology & Novelty:**
  - "Novel task synthesis channel using structured Wikipedia tables ... multi-entity information-seeking tasks." (**vV8B**)
  - "Proposes an innovative tree-based pipeline, ... systematically increases task complexity."1 (**GntM**)
  - "Addresses a neglected but critical dimension—efficiency." (**ZRih**)
  - "The proposed tree-structured reasoning framework can support more comprehensive IS tasks." (**cPZV**)
- **Metrics (ISR & ISE):**
  - "Empirically grounded filtering strategy (ISR/ISE) that directly addresses observed inefficiencies." (**vV8B**)
  - "Clear theoretical justification and measurable impact." (**ZRih**)
  - "The use of ISR and ISE metrics to curate training trajectories is a valuable contribution." (**cPZV**)
- **Evaluation & Results:**
  - "Strong empirical results across five benchmarks, with clear gains and well-supported ablations." (**vV8B**)
  - "Significantly outperform open source baselines, achieving both higher accuracy and greater efficiency."2 (**GntM**)
  - "Highly reproducible (data construction and algorithmic details disclosed)." (**ZRih**)

**Concerns and Our Improvements**

The reviewers expressed concerns regarding the mechanism of efficiency gains, the relationship between our method and RL-based reasoning, the definition of the task scope, and requests for additional ablation studies and model generalizations.

**1. Mechanism, Efficiency vs. Reasoning, and RL Integration**

- **Concerns:**
  - "Limited insight into the underlying mechanism of why the efficiency-oriented filtering leads to such large improvements." (**vV8B**)
  - "The claim that long trajectories are inefficient conflicts with recent RL-based reasoning advances." (**vV8B**)
  - "The ISE filtering may be too rigid, potentially suppressing necessary complex or difficulty-adaptive reasoning." (**vV8B**)
- **Response:**
  - **Clarification:** We clarified the distinction between **Information Seeking** (where "length" often implies redundant actions/noise, making shortness desirable) and **Logical Reasoning** (where "length" implies depth, making shortness risky).
  - **New Experiments** We implemented GRPO on top of our SFT model. The results show that our method is compatible with RL. Crucially, the agent learned to improve performance *without* increasing the number of steps, confirming that for IS tasks, "smarter" is better than "longer."

**2. Generalization (Models, Real-World, & Language)**

- **Concerns:**
  - "Finetuned on a single base model (Qwen3-30B)... observations may change with a different base model." (**GntM**)
  - "May not generalize well to real-world applications... web content is often noisy." (**cPZV**)
  - "No multilingual or real-world deployment tests." (**ZRih**)
  - "Dataset bias (Wikipedia-only) not discussed." (**ZRih**)
- **Response:**
  - **New Experiments**  We conducted experiments using **Qwen3-4B**. The results confirm consistent improvements, proving our method works across model sizes.
  - **Analysis:** We highlighted that our benchmarks (GAIA, BrowseComp, WideSearch) are constituted of noisy, real-world, and multilingual (Chinese/English) data, directly validating robustness beyond Wikipedia.

**3. Task Definition & Structure**

- **Concerns:**
  - "Agent mainly focuses on finding target entities... akin to entity mining tasks rather than general information-seeking tasks." (**cPZV**)
  - "Lacks a detailed analysis of how this structure [Tree] is advantageous compared to other possible structures, such as graphs." (**cPZV**)
- **Response:**
  - **Clarification:** We justified the **entity-centric** definition as a widely adopted foundation for complex reasoning (supported by references like WebResearcher/DeepDive).
  - **Justification:** We explained that the **Tree structure** was chosen specifically to maximize entity density and synthesis scalability, which are the primary bottlenecks we aim to solve.

**4. Ablations & Hyperparameters**

- **Concerns:**
  - "Ablation study could include the analysis of how the average action rounds change." (**GntM**)
  - "Lack of hyperparameter sensitivity analysis for $\alpha$ and $\beta$." (**ZRih**)
- **Response:**
  - **New Analysis:** We added a sensitivity analysis for ISR and ISE thresholds, identifying the "sweet spot" for data quality vs. quantity.
  - **Clarification:** We pointed to Figure 4 analysis which explicitly plots Effectiveness vs. Efficiency (action rounds).

For further details on these experiments and clarifications, we kindly refer the reviewers to our specific responses below.

---

### Author Response · Authors · 2025-11-25
**Revision Summarization**

## Reviewer vV8B

1. **The discussion about why the efficiency-oriented filtering leads to such large improvements**: Line 302-305
2. **WebLeaper with RL training further improves. WebLeaper mainly improves EFFICIENCY and EFFICACY of tool call behaviour**: Appendix A.3
3. **Our core objective in introducing $\mathrm{ISE}$ filtering is to enhance the quality and efficiency of tool calls, not to restrict the length or depth of reasoning content**: Appendix A.4
4. **Discussion on dataset contamination**": Appendix A.11
5. **Clarifications between information-seeking task and entity retrieval**: Appendix A.10



## Reviewer GntM

1. **Permances on different backbones**: Section 4.7



## Reviewer cPZV

1. **Clarifications between information-seeking task and entity mining**: Appendix A.10
2. **Discussion of Performance on Practical Scenario**: Appendix A.11



## Reviewer ZRih

1. **Hyperparameter sensitivity analysis for $\alpha$ and $\beta$**: Section 4.6
2. **Discussion on dataset bias (Wikipedia-only)**: Line 222-225 and Appendix A.11
3. **Discussion of performance on multilingual and practical scenario**: Appendix A.11
4. **Analysis on computational training overhead**: Line 321-348

---

### Author Response · Authors · 2025-11-28
**Clarification of our rebuttal and scores**

## Dear AC, SAC, PC, Reviewers,

Much before this OpenReview bug occurred (02:00 AOE on November 27), we resolved the issues raised by reviewers **vV8B** and **cPZV** through rebuttal.
* **vV8B**'s score was ultimately raised to 8 (19:48 AOE on November 24, 2025) [\[Evidence\]](https://openreview.net/revisions?id=aIl6k2JLYM). Reviewer **vV8B** provided a response and explicitly mentioned in their reply that they raised the score to "8".
* **cPZV**'s score was ultimately raised to 6 (14:20 AOE on November 26) [\[Evidence\]](https://openreview.net/revisions?id=G2qaIeRn9L). Reviewer **cPZV** did not provide any response; they only changed the score.

These score increases were based on our multi-turn conversation and resolution of their issues, which can be verified through the rebuttal history.

**Our paper's final score is 8868 (avg. 7.5, ranking < 100, within 1.8% of all), which is promising to be selected as an Oral paper**. We promise that this score was determined much before the bug occurred.

We wish to ensure that this situation is accurately reflected, and we respectfully request that you take this information into consideration when making your final decision.

Sincerely,

The Authors

---

### Meta-Review · Area_Chair_VPph · 2025-12-29

**Summary:**

This paper identifies low search efficiency as a key limitation of current LLM-based information-seeking agents, largely due to the sparsity of target entities in training tasks. To address this, the paper proposes WebLeaper, a framework for constructing high-coverage IS tasks and generating efficient solution trajectories. It formulates information seeking as a tree-structured reasoning problem and constructs high-coverage tasks using curated Wikipedia tables. By synthesizing diverse task variants and training on trajectories that are both accurate and efficient, WebLeaper consistently improves search effectiveness and efficiency across multiple benchmarks.

The strengths include: 1) the tree-structured framework and the data synthesis method are novel; 2) the focus on efficiency is an important problem; 3) the designed filtering strategy does improve efficiency and accuracy; 4) strong empirical results across multiple benchmarks; and 5) reproducible results. However, the reviewer concerns include:
1. Limited insight, analysis, explanation why the proposed filtering leads to such big improvement (vV8B);
2. The claim that long trajectories are inefficient conflicts with recent RL-based reasoning advances (vV8B);
3. The ISE filtering may be too rigid (vV8B);
4. Missing experiments on other base models (GntM);
5. Missing ablation of how the average action rounds change with different data sources (GntM);
6. Missing some clarification/discussion between the problems of information-seeking and entity finding (cPZV);
7. The concern on the generalization to real-world applications and multilingual scenarios (cPZV, ZRih);
8. Lack of hyperparameter sensitivity analysis for α and β (ZRih);
9. Dataset bias (Wikipedia-only) not discussed (ZRih);
10. Missing analysis on computational training overhead (ZRih).

According to the discussion/review revision before the Openreview bug was released, vV8B and cPZV were happy with the rebuttal and increased their scores to positive. However, for GntM/ZRih's concerns, the AC finds many of the responses are not very complete and satisfying. For example:
- For #4, although the rebuttal provides additional results for Qwen3-4B-thinking-2507, it is in the same family of Qwen3, and their behaviors could be very close to each other. Ideally, some other base models need to be investigated.
- For #5, the rebuttal repeats the observations of Figure 4 of two ablation models. However, the reviewers wanted to see more ablated models.
- For #7, the rebuttal doesn't provide any additional real-world/multilingual test beyond the benchmarks.
- For #8, the rebuttal only provides two data points for α (0.3 and 0) and β (0.1 and 0), which are not enough for convincing sensitivity analysis.
- For #9, still no discussion beyond Wikipedia.

GntM/ZRih gave very positive scores at the beginning (both 8), and the AC thinks the incomplete responses may not necessarily make them change to negative scores. Given the good contributions from this paper, and vV8B/cPZV are happy with the rebuttal, the AC would suggest to accept this paper. But the authors are strongly suggested to complete the responses to GntM/ZRih and take all responses into the final revision.

**Reviewer Concerns:**

The reviewer concerns include:
1. Limited insight, analysis, explanation why the proposed filtering leads to such big improvement (vV8B);
2. The claim that long trajectories are inefficient conflicts with recent RL-based reasoning advances (vV8B);
3. The ISE filtering may be too rigid (vV8B);
4. Missing experiments on other base models (GntM);
5. Missing ablation of how the average action rounds change with different data sources (GntM);
6. Missing some clarification/discussion between the problems of information-seeking and entity finding (cPZV);
7. The concern on the generalization to real-world applications and multilingual scenarios (cPZV, ZRih);
8. Lack of hyperparameter sensitivity analysis for α and β (ZRih);
9. Dataset bias (Wikipedia-only) not discussed (ZRih);
10. Missing analysis on computational training overhead (ZRih).

According to the discussion/review revision before the Openreview bug was released, vV8B and cPZV were happy with the rebuttal and increased their scores to positive. However, for GntM/ZRih's concerns, the AC finds many of the responses are not very complete and satisfying. For example:
- For #4, although the rebuttal provides additional results for Qwen3-4B-thinking-2507, it is in the same family of Qwen3, and their behaviors could be very close to each other. Ideally, some other base models need to be investigated.
- For #5, the rebuttal repeats the observations of Figure 4 of two ablation models. However, the reviewers wanted to see more ablated models.
- For #7, the rebuttal doesn't provide any additional real-world/multilingual test beyond the benchmarks.
- For #8, the rebuttal only provides two data points for α (0.3 and 0) and β (0.1 and 0), which are not enough to do convincing sensitivity analysis.
- For #9, still no discussion beyond Wikipedia.

**Reviewer Scores:**

The original scores are 4 (vV8B), 8 (GntM), 4 (cPZV), 8 (ZRih). According to the discussion/review revision before the Openreview bug was released, vV8B and cPZV updated their scores to 8 and 6, respectively. Although the AC thinks GntM/ZRih's concerns were not well addressed by the rebuttal, the AC doesn't think they will change their scores to negatives.

---

### Decision · Program_Chairs · 2026-01-26

Accept (Poster)